# USP5/Leon deubiquitinase confines postsynaptic growth by maintaining ubiquitin homeostasis through Ubiquilin

Chien-Hsiang Wang[1,2], Yi-Chun Huang[1,3], Pei-Yi Chen[1,2], Ying-Ju Cheng[1], Hsiu-Hua Kao[1], Haiwei Pi[3], Cheng-Ting Chien[1,2]*

[1]Institute of Molecular Biology, Academia Sinica, Taipei, Taiwan; [2]Institute of Neuroscience, National Yang Ming University, Taipei, Taiwan; [3]Department of Biomedical Sciences, Chang Gung University, Taoyuan, Taiwan

**Abstract** Synapse formation and growth are tightly controlled processes. How synaptic growth is terminated after reaching proper size remains unclear. Here, we show that Leon, the *Drosophila* USP5 deubiquitinase, controls postsynaptic growth. In *leon* mutants, postsynaptic specializations of neuromuscular junctions are dramatically expanded, including the subsynaptic reticulum, the postsynaptic density, and the glutamate receptor cluster. Expansion of these postsynaptic features is caused by a disruption of ubiquitin homeostasis with accumulation of free ubiquitin chains and ubiquitinated substrates in the *leon* mutant. Accumulation of Ubiquilin (Ubqn), the ubiquitin receptor whose human homolog ubiquilin 2 is associated with familial amyotrophic lateral sclerosis, also contributes to defects in postsynaptic growth and ubiquitin homeostasis. Importantly, accumulations of postsynaptic proteins cause different aspects of postsynaptic overgrowth in *leon* mutants. Thus, the deubiquitinase Leon maintains ubiquitin homeostasis and proper Ubqn levels, preventing postsynaptic proteins from accumulation to confine postsynaptic growth.

**\*For correspondence:** ctchien@gate.sinica.edu.tw

**Competing interests:** The authors declare that no competing interests exist.

## Introduction

A synapse is a specialized structure where signals are transmitted from a neuron to another neuron or other target cells such as muscles. Proper synapse formation is prerequisite to building functional synapses and constructing neuronal circuits. Synapse abnormalities are suggested to induce neurological and psychological disorders such as autism spectrum disorders and fragile X syndrome (*van Spronsen and Hoogenraad, 2010*). Formation of postsynapses requires coordinated formation of several specialized structures. One prominent postsynaptic feature at neuromuscular junctions (NMJs) is the extensively folded muscular membranes. Specialized folding of postjunctional membranes is thought to increase the area exposed to the synaptic cleft and ensure the effectiveness of neuromuscular transmission (*Sanes and Lichtman, 1999*; *Wu et al., 2010*). In addition to membrane specializations, the postsynaptic density (PSD) is also a common element whose size requires proper control. The PSD contains scaffolding proteins that recruit signaling protein complexes and neurotransmitter receptors, matching precisely the presynaptic active zones (*Feng and Zhang, 2009*; *Sheng and Hoogenraad, 2007*). Formations of postsynaptic membrane and PSD are tightly controlled and coordinated yet these processes remain elusive.

The *Drosophila* NMJ is a model to study synapse formation and activity-dependent synapse remodeling (*Collins and DiAntonio, 2007*; *Ruiz-Cañada and Budnik, 2006*). Synaptic boutons are swollen structures of axonal terminals embedded in highly folded muscular membranes called the subsynaptic reticulum (SSR) and each bouton contains tens of neurotransmitter release sites paired with PSDs. During larval development, the SSR and the PSD concomitantly form and gradually

increase their sizes. Two crucial factors, postsynaptic density protein-95/Discs large (Dlg) localized at the SSR and *Drosophila* p21-activating kinase (dPak) localized at the PSD, regulate SSR formation (*Albin and Davis, 2004*; *Budnik et al., 1996*; *Lahey et al., 1994*). At the PSD, two types of localized glutamate receptors (GluRs), IIA and IIB, appear in distinct GluR clusters (*Marrus et al., 2004*). The abundance of GluRIIA at the PSD is regulated by PSD-localized dPak and the SSR-localized NF-κB complex, NF-κB/Dorsal (Dl), IκB/Cactus (Cact) and IRAK/Pelle (Pll) (*Albin and Davis, 2004*; *Heckscher et al., 2007*; *Parnas et al., 2001*; *Zhou et al., 2015*). Thus, the postsynaptic protein could localize at either SSR or PSD, and confer growth regulation on SSR, PSD or both.

Ubiquitination plays essential roles in various cellular processes including synaptic growth (*DiAntonio et al., 2001*). Ubiquitin species are dynamically balanced among free and substrate-con-jugated forms of mono-ubiquitin and ubiquitin chains. Ubiquitin homeostasis, i.e. the maintenance of diverse ubiquitin species in proper proportions and levels, is regulated in cellular growth and dif-ferentiation (*Hallengren et al., 2013*; *Kimura and Tanaka, 2010*). Deubiquitinases (DUBs), a large superfamily of ubiquitin regulators, participate in the dynamic equilibrium of ubiquitin species. While some DUBs process newly synthesized ubiquitin precursors for ubiquitin supply, others recycle ubiq-uitin by cleaving ubiquitin chains from protein substrates prior to proteasomal degradation. USP5, the focus of this study, is dedicated to disassembly of free ubiquitin chains for recycling (*Hoch-strasser, 2009*; *Komander et al., 2009*). Physiologically, heat shock stress in yeast causes a reduc-tion of the mono-ubiquitin level. To compensate for ubiquitin depletion, the level of the DUB Doa4 is elevated, leading to an increase in the mono-ubiquitin level by cleaving free ubiquitin chains (*Kimura et al., 2009*). The ataxia mice $ax^J$, carrying mutations in the DUB USP14, displayed nerve swelling and abnormal neurotransmission at NMJs. The defects are caused by a reduction in the ubiquitin level as lower ubiquitin levels were detected in the mutant mice and introducing an ubiqui-tin transgene suppressed the $ax^J$ phenotypes (*Chen et al., 2011*, *2009*). Thus, regulation of the ubiquitin level is a critical step in synapse development and for preventing neurological disorders.

*Drosophila* USP5/Leon is essential to maintain ubiquitin homeostasis during tissue formation and controls activation of apoptosis and the JNK pathway during eye development (*Fan et al., 2014*; *Wang et al., 2014*). In this study, we characterized the role of Leon in postsynaptic growth after syn-apse formation. In *leon* mutants, while the presynapse maintains normal morphology, the postsy-napse overelaborates, displaying expanded SSR, enlarged PSD and excess PSD-localized GluR clusters. Free ubiquitin chains and ubiquitinated substrates accumulate in *leon* postsynapses, reveal-ing defects in ubiquitin homeostasis. Genetic analysis shows that accumulations of several postsyn-aptic proteins accounts for overelaborated postsynaptic structures. The ubiquitin receptor Ubqn recognizes and transfers ubiquitinated substrates to the proteasome for degradation (*Finley, 2009*; *Lipinszki et al., 2011*). The Ubqn level is elevated in *leon* postsynapses and reducing the Ubqn level suppresses *leon* mutant phenotypes. Importantly, co-overexpression of free ubiquitin chains and Ubqn promotes expansion of these postsynaptic features. Thus, ubiquitin homeostasis such as disas-sembly of free ubiquitin chains, timely degradation of proteins, and normal function of the ubiquitin receptor Ubqn are compromised in *leon* mutants, leading to postsynaptic overgrowth.

## Results

### Abnormal NMJ morphology in *leon* mutants

We examined NMJs in third-instar larvae of $leon^2/19\text{-}2$ mutants that died at mid-pupal stages and $leon^1/19\text{-}2$ mutants that died at the late-third larval stages. Based on the viability and Western blot analysis, $leon^2/19\text{-}2$ is considered a hypomorphic mutant and $leon^1/19\text{-}2$ is a close-to-null mutant (*Wang et al., 2014*). Both NMJ phenotypes were compared to the control $w^{1118}$ that had been used to backcross all *leon* alleles. At control NMJs, axonal terminals immunostained for presynaptic horse-radish peroxidase (HRP) branched out extensively from initial targeting sites; synaptic boutons revealed by postsynaptic Dlg staining spread evenly along axonal tracks, displaying the beads-on-a-string pattern (*Figure 1A*). In *leon* mutants, axonal terminals failed to extend and synaptic boutons aggregated, making them larger in appearance (*Figure 1A*). To quantify the morphological defects, we scored the number of boutons and branch lengths of NMJs at muscles 6/7. In both $leon^1/19\text{-}2$ and $leon^2/19\text{-}2$ mutants, the bouton numbers were significantly decreased by about 25% (*Figure 1B*). Total branch length was also reduced in $leon^1/19\text{-}2$, although the reduction was not

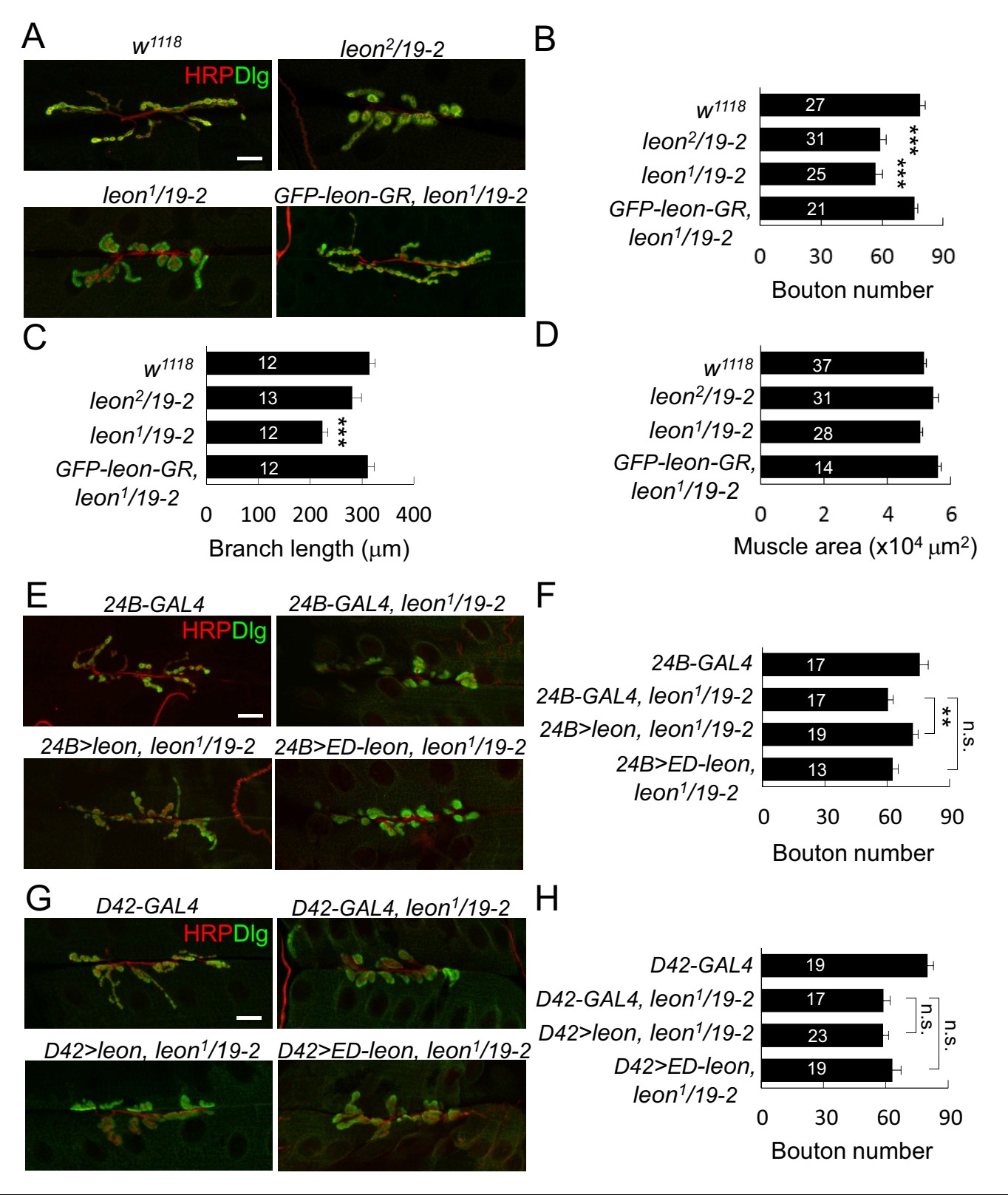

**Figure 1.** *leon* mutants display abnormal NMJ morphology. (**A**) Immunostaining images show NMJs of $w^{1118}$, $leon^2$/19-2, $leon^1$/19-2 and *GFP-leon-GR*, $leon^1$/19-2 for HRP (red) and Dlg (green). (**B–D**) Bar graphs show means ± SEM (standard error of mean) of bouton numbers (**B**), branch lengths (**C**) and muscle areas (**D**, quantified by phalloidin staining, not shown). (**E** and **G**) Immunostaining images co-stained by HRP (red) and Dlg (green) show

*Figure 1 continued on next page*

*Figure 1 continued*

transgene rescue of *leon¹/19-2* NMJs by *UAS-leon* or *UAS-ED-leon* driven by postsynaptic *24B-GAL4* (E) or presynaptic *D42-GAL4* (G). (F and H) Bar graphs show means ± SEM of bouton numbers in postsynaptic or presynaptic rescue. Scale bars, 20 μm. All data were compared to controls unless specifically indicated by brackets, with n.s. indicating no significance, * for p<0.05, and *** for p<0.001 according to Student's *t* tests. The detail statistic numbers also see ***Supplementary file 1***.

The following figure supplement is available for figure 1:

**Figure supplement 1.** Postsynaptic *leon* in NMJ morphology and Leon expression pattern.

significant in *leon²/19-2* (***Figure 1C***). In both mutants, the muscle sizes were comparable to controls (***Figure 1D***). Reductions in both bouton number and branch length in *leon¹/19-2* were completely restored by *GFP-leon-GR*, a genomic rescue transgene of *leon,* suggesting that *leon* is required for normal NMJ formation (***Figure 1A–C***).

We then analyzed whether *leon* is required in pre- or post-synapses by performing tissue-specific rescue of *leon* mutants. Expression of *UAS-leon* by *24B-GAL4* in muscles of *leon¹/19-2* restored NMJ morphology and bouton number to the wild-type level (***Figure 1E,F***). In contrast, expression by *D42-GAL4* in motor neurons failed to rescue *leon* mutant phenotypes (***Figure 1G,H***). The post-synaptic requirement of *leon* was further confirmed by driving the *leonRNAi* transgene in individual compartments. When driven by *24B-GAL4,* the NMJ morphological defect was identical to that of *leon* mutants, with a significant reduction in the bouton number (***Figure 1—figure supplement 1A, C***). These phenotypes, however, were not detected by neuronal expression of *leonRNAi* (***Figure 1—figure supplement 1B,C***).

To analyze the requirement of Leon deubiquitinating activity for NMJ growth, the *UAS-ED-leon* transgene that expresses enzyme-dead Leon was introduced into *leon* mutants. Expression of *UAS-ED-leon* in muscles or motor neurons failed to rescue any of the *leon* mutant phenotypes (***Figure 1E–H***). Thus, in comparison to the effective rescue by wild-type *UAS-leon*, this result suggests that Leon functions as a DUB in regulating NMJ development.

Immunostaining of larval tissues by anti-Leon antibodies showed that Leon was expressed ubiquitously. At NMJs, Leon was enriched within synaptic boutons (***Figure 1—figure supplement 1D***, arrowheads). Leon was also expressed in postsynapses with lower levels in the SSR marked by Cact (***Figure 1—figure supplement 1E***). These Leon expressions were almost diminished in the null *19–2* homozygous larvae, confirming that these signals represent Leon expression (***Figure 1—figure supplement 1D***, bottom panels). Residual puncta (arrowheads) within presynaptic boutons may represent background signals. The genomic rescue transgene *GFP-leon-GR* showed a similar expression pattern to endogenous Leon (***Figure 1—figure supplement 1F***), further confirming Leon expression at the NMJ.

## Defective ubiquitin homeostasis in *leon* postsynapses

To examine whether ubiquitin homeostasis is disrupted in the *leon* mutant, we first performed Western blots to analyze ubiquitin profiles in dissected body wall muscles. By blotting with the anti-ubiquitin antibodies, we found that the levels of free ubiquitin chains were increased in *leon* mutants (***Figure 2A***). Whereas moderate increases were detected in hypomorphic *leon²/19-2* (2.33 ± 0.64 folds, see Materials and methods), large increases were found in null *19-2/19-2* and close-to-null *leon¹/19-2* (4.25 ± 1.38 and 4.62 ± 1.61 folds, respectively). In addition, smearing signals at higher molecular weights representing ubiquitinated substrates were also increased in *19-2/19-2* and *leon¹/19-2* (1.88 ± 0.31 and 1.76 ± 0.37 folds, respectively). The higher molecular-weight smears, however, were increased slightly in *leon²/19-2* (1.29 ± 0.78 folds). In contrast, the level of monoubiquitin was strongly increased in hypomorphic *leon²/19-2* (1.9 ± 0.77 folds) but the increases were not prominent in *19-2/19-2* and *leon¹/19-2* (0.98 ± 0.17 and 1.25 ± 0.2 folds, respectively). Therefore, the severity in defective ubiquitin homeostasis, represented by the increased levels of free ubiquitin chains and ubiquitinated substrates, correlates with increasing *leon* mutant strengths.

To further confirm disrupted ubiquitin homeostasis in *leon* mutants, the FK2 antibody that recognizes mono- and poly-ubiquitinated substrates, and free ubiquitin chains was used to immunostain dissected larval body walls. In wild-type controls, FK2 signals were detected at NMJs and in nuclei

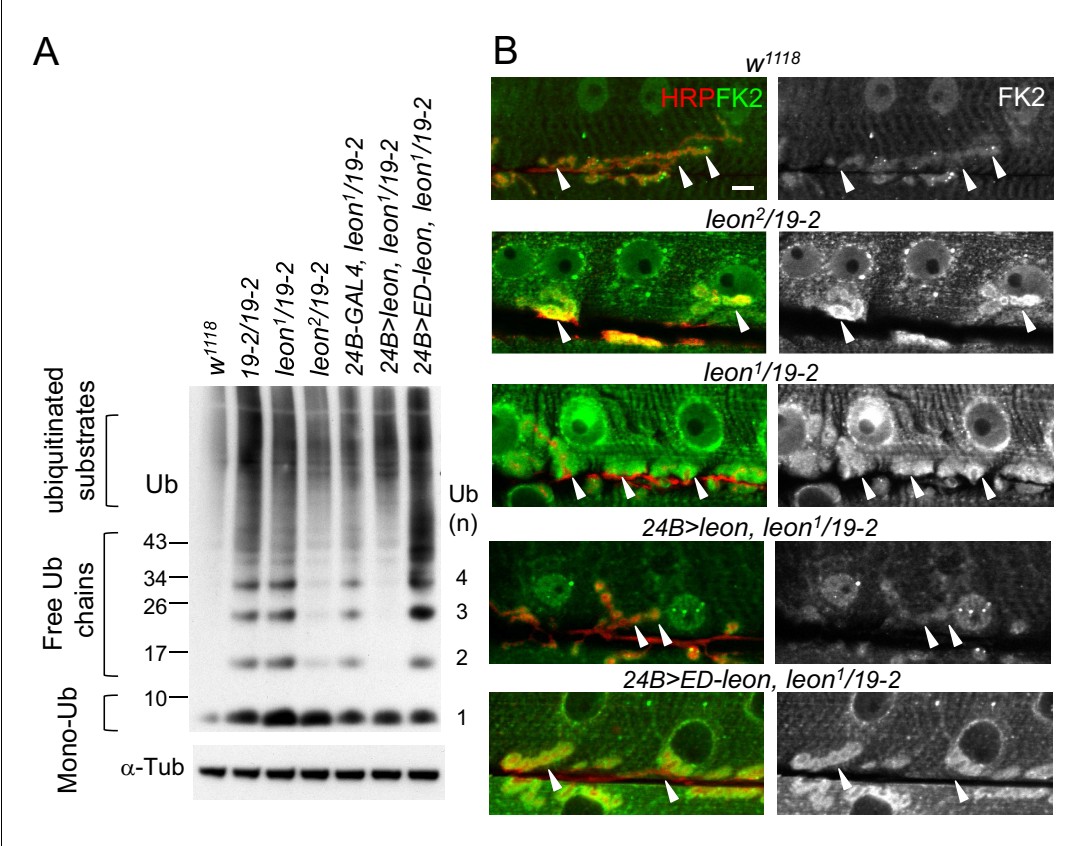

**Figure 2.** Defective ubiquitin homeostasis in *leon* mutants. (**A**) Western blot probed with ubiquitin (Ub) antibodies shows ubiquitin expression patterns in *w1118*, *19-2/19-2*, *leon1/19-2*, *leon2/19-2*, and postsynaptic expression of *UAS-leon* or *UAS-ED-leon* driven by *24B-GAL4* in *leon1/19-2*. α-Tub as control. (**B**) Images show FK2 (green) and HRP (red) immunostaining of NMJs in *w1118*, *leon2/19-2* and *leon1/19-2*, and postsynaptic expression of *UAS-leon* or *UAS-ED-leon* driven by *24B-GAL4* in *leon1/19-2*. The single FK2 images (right) are also shown. Arrowheads indicate synaptic regions. Scale bar, 10 μm.

and Z-bands of muscles (*Figure 2B*). In *leon* mutants, overall FK2 signals were enhanced, and the enhancement was very prominent at the postsynaptic sites surrounding presynaptic boutons (indicated by arrowheads, *Figure 2B*). Consistent with Western blot analysis, the enhancement in the FK2 signal level was more pronounced in *leon1/19-2* than in *leon2/19-2* (*Figure 2B*).

We then examined the requirement of postsynaptic Leon and its deubiquitinating activity for ubiquitin homeostasis. In *leon1/19-2*, muscle expression of *UAS-leon*, but not *UAS-ED-leon*, by *24B-GAL4* suppressed the enhanced FK2 signals (*Figure 2B*). Western blot analysis showed that *24B-GAL4*-driven *UAS-leon* expression suppressed elevated free ubiquitin chains in *leon1/19-2* (0.49 ± 0.02 folds in comparison to *24B-GAL4, leon1/19-2*) and partially suppressed elevated ubiquitinated substrates (0.91 ± 0.04 folds). The partial suppression could be attributed to the limited muscle expression in the mutant. Interestingly, the expression of *UAS-ED-Leon* induced higher levels of free ubiquitin chains (2.17 ± 0.16 folds) and ubiquitinated substrates (2.19 ± 0.43 folds). It is possible that the enzymatic activity-deficient ED-Leon can associate with ubiquitin chains but cannot deubiquitinate them, and binding to ED-Leon sequesters ubiquitinated substrates from degradation. Taken together, these results indicate that Leon deubiquitinating activity is required for maintaining ubiquitin homeostasis in postsynapses.

## Increases of postsynaptic proteins in *leon* mutants

Aberrant NMJ morphology and postsynaptic ubiquitin homeostasis defects in *leon* mutants prompted us to examine constituents of synaptic organization. Several pre- and post-synaptic proteins were analyzed for their expression patterns and levels in *leon* mutants. Immunostaining for Dlg

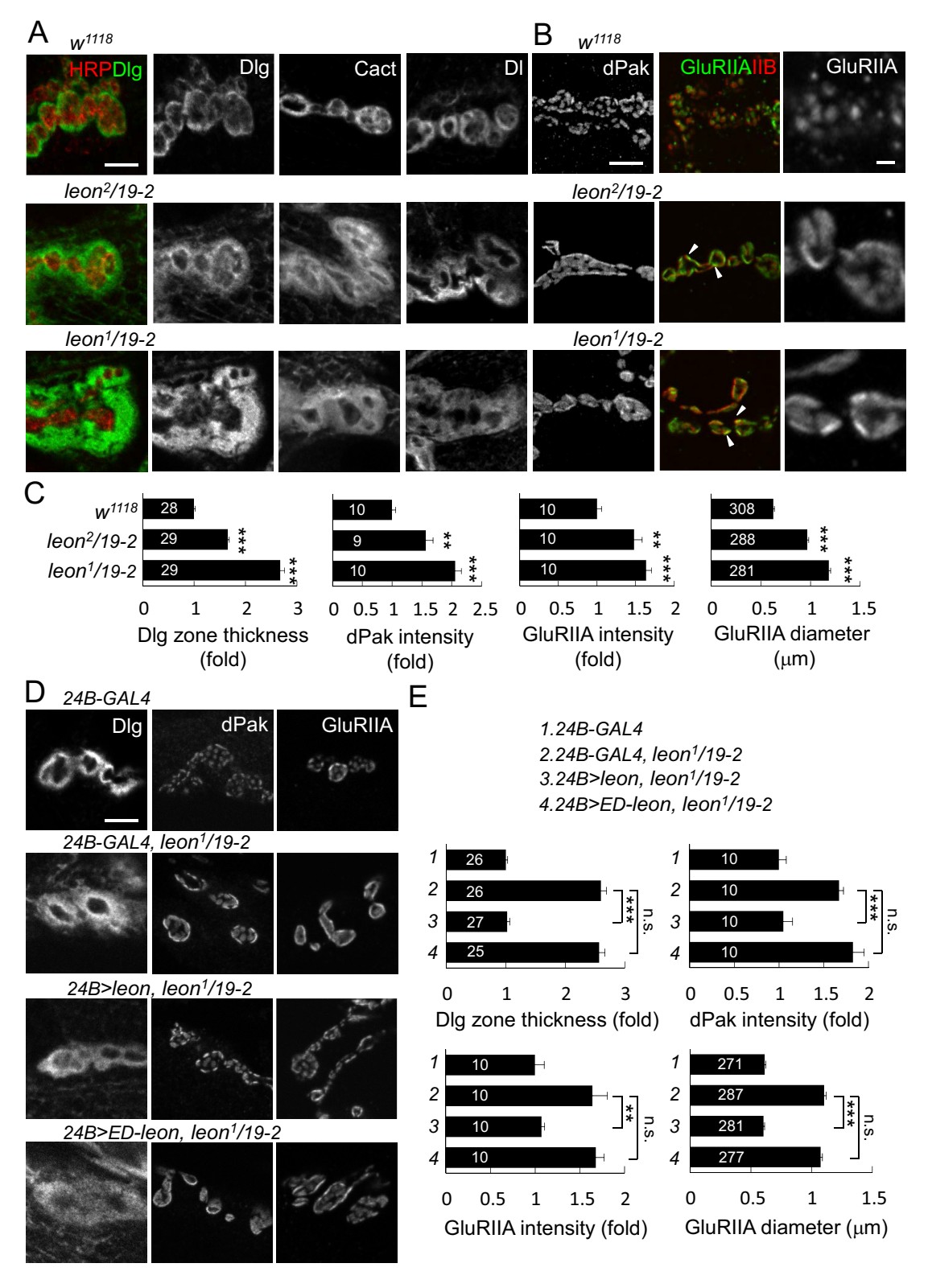

**Figure 3.** Expansion of SSR- and PSD-localized proteins at *leon* NMJs. (**A**) Images show SSR-localized proteins Dlg, Cact and Dl in immunostaining of boutons in $w^{1118}$, $leon^2/19$-2 and $leon^1/19$-2. Co-staining of Dlg (green) and HRP (red) is shown (left panels). (**B**) Images show PSD-localized dPak, GluRIIA and GluRIIB immunostaining at synapses of $w^{1118}$, $leon^2/19$-2 and $leon^1/19$-2. Arrowheads indicate overlapping signals of expanded GluRIIA (green) and GluRIIB (red) clusters in middle panels. GluRIIA images are shown with magnification (right panels). (**C**) Bar graphs show means ± SEM of

*Figure 3 continued on next page*

*Figure 3 continued*

Dlg-positive zone thickness, dPak intensity and GluRIIA intensity and diameter in $w^{1118}$, $leon^2/19\text{-}2$ and $leon^1/19\text{-}2$. (**D**) Images show Dlg, dPak and GluRIIA immunostaining at NMJs, with postsynaptic *24B-GAL4*-driven *UAS-leon* or *UAS-ED-leon* expression in $leon^1/19\text{-}2$. (**E**) Bar graphs show means ± SEM of Dlg-positive zone thickness, dPak intensity and GluRIIA intensity and diameter. All scale bars represent 5 µm except in magnified GluRIIA images, which is 1 µm. All data were compared to controls unless specifically indicated by brackets with n.s. indicating no significance, ** for p<0.01 and *** for p<0.001 according to Student's *t* tests. The detail statistic numbers also see ***Supplementary file 1***.

The following figure supplements are available for figure 3:

**Figure supplement 1.** Localization of presynaptic proteins and expansion of GluRIIC clusters at *leon* NMJs.

**Figure supplement 2.** Postsynaptic defects of *leon* mutants in earlier larval stages.

**Figure supplement 3.** Presynaptic *leon* is not required for postsynaptic formation.

**Figure supplement 4.** Postsynaptic *leon* knockdown exhibits postsynaptic defects.

that is enriched in the SSR revealed a striking phenotype. In controls, SSR-localized Dlg exhibited thin circular rings in bouton sections. In *leon* mutants, Dlg-localized rings were expanded, showing much greater thickness and higher protein levels (***Figure 3A***). The thickness of Dlg-positive zones increased by 66% in $leon^2/19\text{-}2$ and by almost two folds in $leon^1/19\text{-}2$ in comparison to controls (***Figure 3C***, left panel). We further examined SSR-localized Cact and Dl expressions, which show Dlg-resembling ring patterns (***Heckscher et al., 2007***). In *leon* mutants, circular Cact and Dl patterns were also expanded (***Figure 3A***). As Dlg, Cact and Dl present different aspects of postsynaptic functions, with Dlg promoting SSR formation, and Cact and Dl regulating GluRIIA abundance (***Budnik et al., 1996***; ***Heckscher et al., 2007***; ***Lahey et al., 1994***), expansions of SSR-localized Dlg, Cact and Dl suggest that the SSR is also likely expanded in *leon* mutants.

At NMJs, each bouton contains multiple release sites paired with discrete receptor clusters that can be revealed by the localizations of presynaptic ELKS/CAST family protein Bruchpilot (Brp), and postsynaptic dPak (***Albin and Davis, 2004***; ***Kittel et al., 2006***; ***Wagh et al., 2006***). Well-matched pairs of Brp and dPak were evenly distributed along the rim of and within bouton sections (***Figure 3—figure supplement 1A***). In *leon* mutants, the size, spacing and density of Brp puncta appeared normal (***Figure 3—figure supplement 1A–B***). Strikingly, the postsynaptic dPak patches were enormously enlarged, and the spacing among them was often diminished, appearing as a continuous structure in bouton sections, particularly in $leon^1/19\text{-}2$ (***Figure 3B***). Quantification showed that the dPak levels were increased in *leon* postsynapses (***Figure 3C***). The normal one-to-one pairing between Brp and dPak could not be resolved in *leon* mutants because expanded and fused dPak patches could accommodate a few Brp puncta.

We further examined PSD-localized GluRs. GluRs are composed of four subunits, including essential subunits GluRIIC/GluRIII, GluRIID and GluRIIE and one of the two interchangeable subunits GluRIIA and GluRIIB (***DiAntonio, 2006***; ***Marrus et al., 2004***). The GluRIIA and GluRIIB receptor clusters were distributed evenly in wild-type boutons. In *leon* mutants, GluRIIA and GluRIIB clusters were enlarged and overlapped, filling most of the space in bouton sections (***Figure 3B***, arrowheads). The wild-type GluRIIA cluster pattern, shown in the magnified image, was no longer present in *leon* mutants. Instead, GluRIIA clusters also appeared much like the dPak fusion pattern (***Figure 3B***, right panels). The average size of GluRIIA clusters was increased by 54% in $leon^2/19\text{-}2\%$ and 89% in $leon^1/19\text{-}2$ (***Figure 3C***). The GluRIIC clusters were also enlarged in *leon* mutants (***Figure 3—figure supplement 1A***). Statistically, the intensities of GluRIIA, GluRIIB and GluRIIC immunostaining were all elevated in both *leon* mutants (***Figure 3C*** and ***Figure 3—figure supplement 1B***). These results indicate that the levels of postsynaptic SSR- and PSD-localized proteins are increased in *leon* mutants.

GluR clustering appears after axonal terminals innervate muscles in mid embryonic stages and SSR is formed in the first instar stage (***Guan et al., 1996***; ***Harris and Littleton, 2015***). The increases of SSR- and PSD-localized proteins in *leon* mutants could be a cumulative process from early larval stages. We compared expressions of SSR-localized Cact and PSD-localized GluRIIA in controls and *leon* mutants in the same stages. At *leon* NMJs, the thickness of Cact-positive zones and the

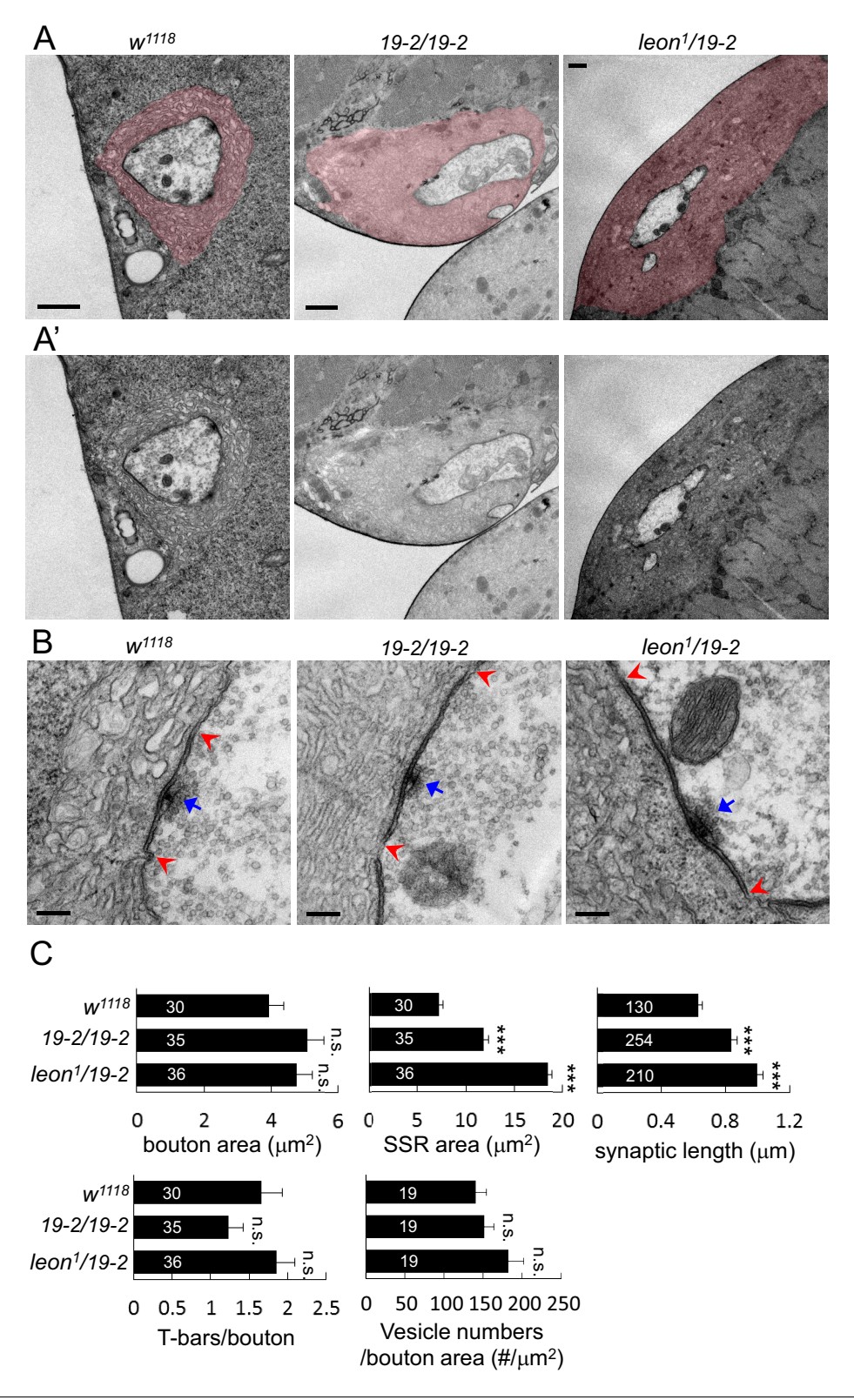

**Figure 4.** Ultrastructural analysis of *leon* mutant boutons showed enlarged SSR and longer synaptic membrane. (**A** and **A'**) Electron micrographs of type Ib boutons in *w*[1118], *19-2/19-2* and *leon*[1]*/19-2*. The SSRs are colored in (**A**). Scale bars, 1 μm. (**B**) Electron-dense membranes with presynaptic T-bars, postsynaptic SSRs and vesicles are shown in *w*[1118], *19-2/19-2* and *leon*[1]*/19-2*. Each pair of red arrowheads delineates the boundary of synaptic membranes, and arrows (blue) indicate T-bars. Scale bars, 0.2 μm. (**C**) Bar graphs show means ± SEM of bouton area, SSR area, synaptic length, T-bar/
*Figure 4 continued on next page*

*Figure 4 continued*

bouton and vesicles/bouton areas in $w^{1118}$, *19-2/19-2* and *leon$^1$/19-2*. All data were compared to $w^{1118}$ with n.s. indicating no significance and *** for p<0.001 by Student's *t* tests. The detail statistic numbers also see *Supplementary file 1*.

intensity and size of GluRIIA clusters were larger than wild-type controls during 72 hr, 96 hr, and 120 hr AEL, with more severe defects in *leon$^1$/19-2* than in *leon$^2$/19-2* (*Figure 3—figure supplement 2A–C*). Therefore, the increases of SSR- and PSD-localized postsynaptic proteins in *leon* mutants are already prominent in early larval stages and progressively enhanced throughout later larval stages.

We then examined whether Leon deubiquitinating activity is required for proper postsynaptic growth. Expression of *UAS-leon* in muscles suppressed the expanded Dlg rings and the enlarged dPak and GluRIIA clusters in *leon$^1$/19-2*. In contrast, enzyme-dead *UAS-ED-leon* failed to suppress these phenotypes (*Figure 3D–E*). Also, presynaptic expression of either *UAS-leon* or *UAS-ED-leon* had no effect on the increased size or intensity of the Dlg-positive zone, the dPak patch and the GluRIIA cluster in *leon* mutants (*Figure 3—figure supplement 3A–B*). To further confirm the requirement of postsynaptic *leon* for proper control of SSR- and PSD-localized protein levels, *leonRNAi* knockdown was performed in pre- or post-synaptic sites. When the *leonRNAi* transgene was driven by postsynaptic *24B-GAL4,* the Dlg-positive zone, the dPak patch, and the GluRIIA cluster were increased (*Figure 3—figure supplement 4A,C*). These phenotypes, however, were not detected by presynaptic knockdown in neurons (*Figure 3—figure supplement 4B,C*). Taken together, these analyses suggest that the deubiquitinating activity of Leon is required in postsynapses to control postsynaptic protein levels.

With the normal distribution, size and intensity of Brp puncta in *leon* mutants, we examined expression patterns of other presynaptic proteins. Immunostaining for synaptic vesicle-associated Synapsin and cysteine string protein (CSP) that display a pattern of small spreading puncta in boutons (*Fuentes-Medel et al., 2009*), revealed no abnormality in *leon* mutants. The signals of microtubule-associated Futsch staining reach the terminal boutons in wild-type larvae, but stopped short without reaching terminals in *leon* mutants, suggesting a possible defect in the microtubule structure or stability in presynapses (*Figure 3—figure supplement 1C*).

## Enlarged SSRs and synaptic membranes in *leon* mutants

We further examined the ultrastructures of *leon* mutant boutons by transmission electron microscopy. Folded SSR surrounding boutons (colored in pink in *Figure 4A*), electron-dense membranes (within the pairs of red arrows in *Figure 4B*) and T-bars (indicated by blue arrows) were analyzed in *19-2/19-2* and *leon$^1$/19-2*. In both mutants, SSR areas were expanded and membrane folds were tightly packed, resulting in compact membrane layers surrounding the sectioned bouton (*Figure 4A, A'*). While the bouton sizes in both *leon* mutants were comparable to controls, the SSR areas were dramatically increased by as much as two-fold (*Figure 4C*). Thus, the ultrastructural analysis confirms the expansion of SSR suggested by analyzing the SSR-localized Dlg, Cact and Dl in *leon* mutants (*Figure 3A*).

The electron-dense synaptic membranes along the bouton circumference were also prominently increased in *leon* mutants (*Figure 4B*). The average length of synaptic membranes was significantly increased in both *leon* mutants (*Figure 4C*). Therefore, synaptic membranes account for more than 60% of the bouton circumference in *leon* mutants, as compared to about 35% in controls. This phenotype is consistent with the enlargement of the PSD that is suggested by immunostaining of dPak and GluRs (*Figure 3B*). Presynaptically, the number of T-bars and vesicle numbers showed no significant difference to controls (*Figure 4C*). Thus, the ultrastructural analysis confirms the expansion of SSR and PSD in *leon* mutants.

## Impaired electrophysiological properties at *leon* mutant NMJs

We then examined the electrophysiological properties in *leon* mutants because of abnormal NMJ morphology in *leon* mutants. In both *19-2/19-2* and *leon$^1$/19-2* mutants, the mEJC frequencies were comparable to the *+/19–2* control, showing no significant differences (*Figure 5A*). mEJC amplitudes in *19-2/19-2* and *leon$^1$/19-2* were slightly larger than that in *+/19–2* control, although no statistical significance was detected (p=0.07 and p=0.13, respectively). Another independent mEJC

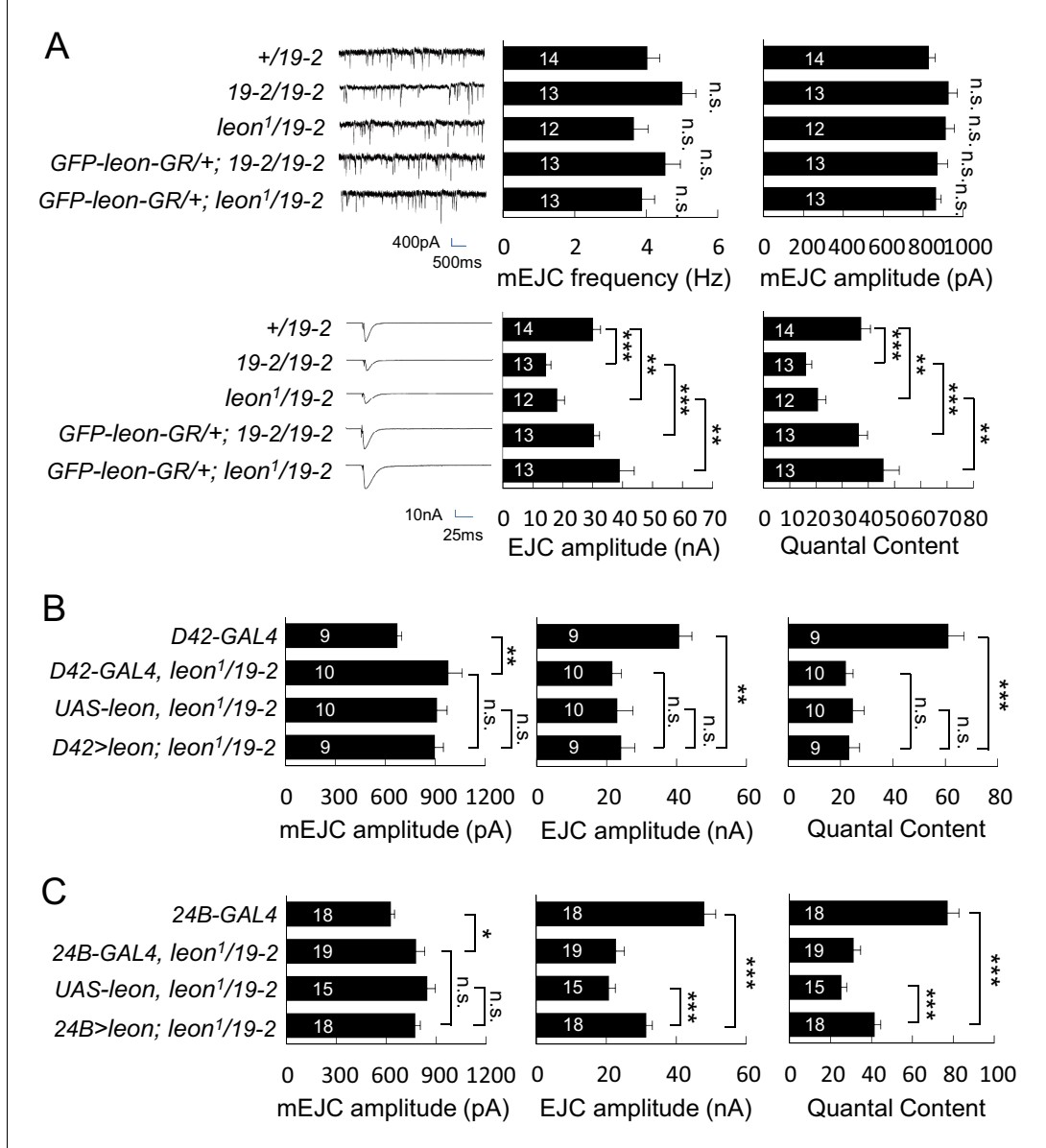

**Figure 5.** Impaired electrophysiological properties at *leon* mutant NMJs. (**A**) Bar graphs compare frequency, mEJC amplitude, EJC amplitude, and quantal content in *19-2/19-2* and *leon[1]/19-2* to *+/19–2* control, and the rescue of *19-2/19-2* and *leon[1]/19-2* by *GFP-leon-GR*. (**B**) Bar graphs show that presynaptic *D42-GAL4*-driven Leon expression failed to restore both EJC amplitude and quantal content in *leon[1]/19-2* mutants. (**C**) Bar graphs show that *24B-GAL4*-driven muscular expression of *leon* partially restored EJC amplitude and quantal content in *leon[1]/19-2*. All data were compared to controls unless specifically indicated by brackets with n.s. indicating no significance, * for p<0.05, ** for p<0.01 and *** for p<0.001 by Student's *t* tests. The detail statistic numbers are in ***Supplementary file 1***.

The following figure supplement is available for figure 5:

**Figure supplement 1.** mEJC amplitude and failure analysis.

amplitudes recording also suggested *leon[1]/19-2* had slightly larger mEJC amplitudes than *+/19–2* and *w[1118]* controls (***Figure 5—figure supplement 1A***). Interestingly, the EJC showed a dramatic reduction in *leon* mutants compared to the *+/19–2* control. The reduction in EJC leads to a reduction in the quantal content, determined by the ratio of the EJC amplitude to the mEJC amplitude (***Figure 5A***, bottom panels). Failure analysis also suggests that the quantal content was significantly

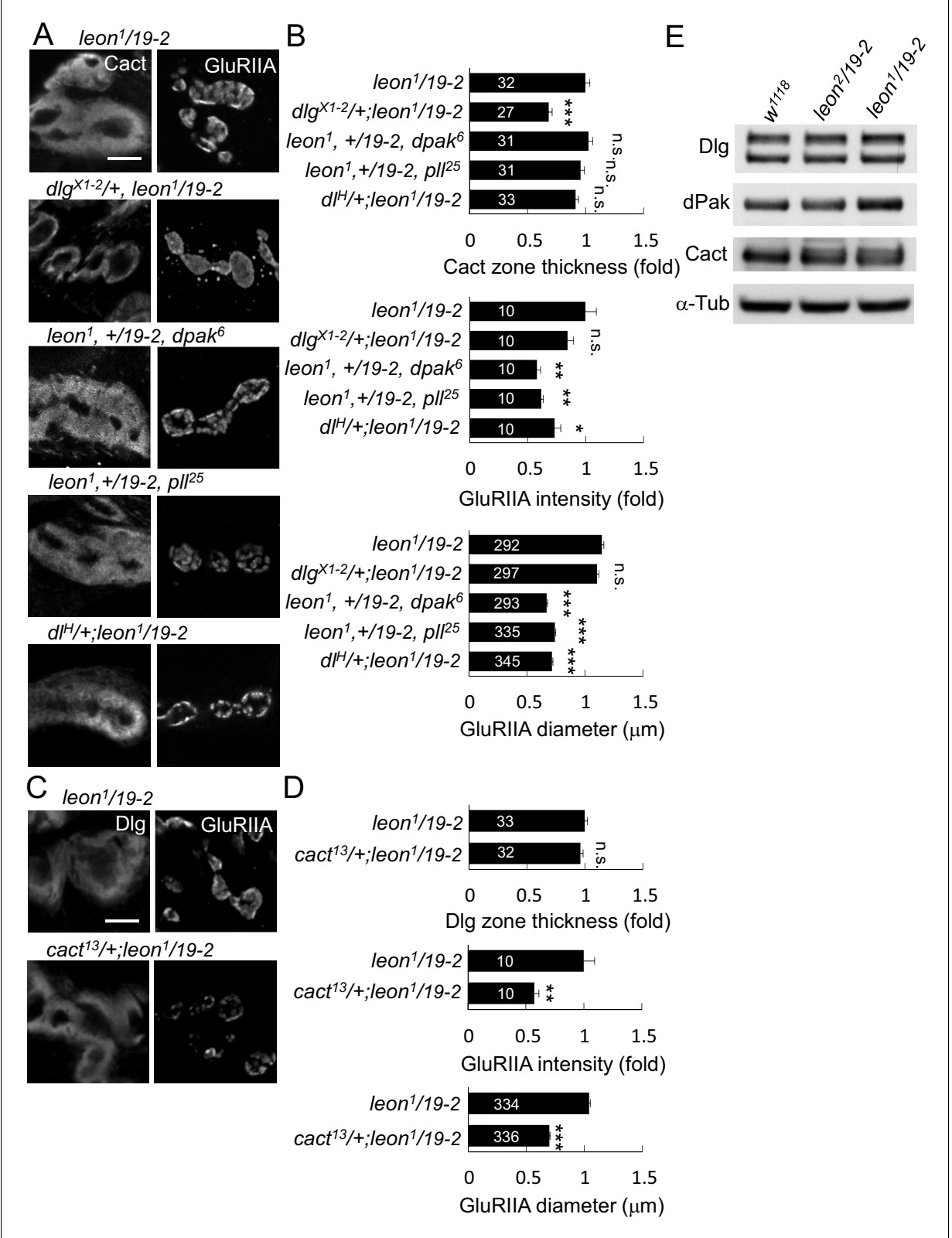

**Figure 6.** *leon* mutant phenotypes suppressed by reductions of gene dosages. (**A**) Immunostaining images show suppression of Cact expansion in *leon¹/19-2* by *dlgˣ¹⁻²*, and suppression of GluRIIA enlargement by *dpak⁶*, *pll²⁵* and *dlᴴ*. (**B**) Bar graphs show means ± SEM of Cact-positive zone thickness and GluRIIA intensity and diameter. (**C**) Immunostaining images of NMJs for Dlg and GluRIIA show suppression of GluRIIA enlargement in *cact¹³/+; leon¹/19-2* as compared to *leon¹/19-2*. (**D**) Bar graphs show means ± SEM of Dlg-positive zone thickness and GluRIIA intensity and diameter. *Figure 6 continued on next page*

*Figure 6 continued*

Comparisons to leon[1]/19-2 were assessed by Student's *t* tests with n.s. indicating no significance, * for p<0.05, ** for p<0.01 and *** for p<0.001. Scale bar, 5 μm. The detail statistic numbers are in ***Supplementary file 1***. (**E**) Western blots show enhanced signals of Dlg, dPak and Cact in leon[1]/19-2 in comparison to w[1118]. α-Tub as control.

The following figure supplement is available for figure 6:

**Figure supplement 1.** *leon* mutant phenotypes suppressed by knockdown of postsynaptic proteins.

reduced in the leon[1]/19-2 mutant (***Figure 5—figure supplement 1B***). Both EJC amplitude and quantal content were restored by introducing *GFP-leon-GR* into *leon* mutants (***Figure 5A***).

We then examined whether Leon is required in pre- or post-synapses for restoring the EJC and quantal content in leon[1]/19-2. Presynaptic Leon expression by *D42-GAL4* failed to restore both EJC amplitude and quantal content in *leon* mutants (***Figure 5B***). Postsynaptic Leon expression by *24B-GAL4* partially restored the EJC amplitude and the quantal content in leon[1]/19-2 (***Figure 5C***). The mEJC in *leon* mutants carrying *D42-GAL4* or *24B-GAL4* was significantly larger than that in respective GAL4 driver control, suggesting the mEJC increase in *leon* mutants might be sensitive to the variation in genetic backgrounds (***Figure 5B,C***). As postsynaptic expression of Leon also suppressed bouton reduction in the *leon* mutant, the reduction of the bouton and hence the total release sites in *leon* mutants could contribute partly to the electrophysiological defects. Indeed, there were about 20% reduction in the bouton number (***Figure 1A,B,E and F***), and about 15% reduction in the release sites at the *leon* mutant NMJ (Brp number per NMJ: w[1118], 731.6 ± 31.6; leon[2]/19-2, 631.9 ± 28.1; leon[1]/19-2, 619.5 ± 31.1; n = 10 for all genotypes). With the 50% or more reduction in the EJC amplitude and the quantal content, other processes are also likely defective in the *leon* mutants.

## Suppression of *leon* mutant phenotypes by reducing postsynaptic proteins

Accumulations of postsynaptic proteins in the SSR or the PSD of *leon* mutants might cause some aspects of the mutant phenotypes. To test this idea, we examined whether reductions of the gene dosage for these postsynaptic proteins would alleviate *leon* mutant phenotypes. We first examined SSR-localized Dlg that is required for SSR formation (***Lahey et al., 1994***). Replacing the wild-type allele by the null dlg[X1-2] allele significantly suppressed the SSR-localized Cact expansion in leon[1]/19-2 (***Figure 6A,B***). However, the intensity and size of PSD-localized GluRIIA clusters was unaltered. Thus, this result suggests that the Dlg might mediate SSR but not PSD expansion in the *leon* mutant.

The PSD-localized dPak is required for GluRIIA cluster localization and SSR formation (***Albin and Davis, 2004***). Replacing the wild-type copy of *dpak* by the null dpak[6] allele in leon[1]/19-2 suppressed the GluRIIA cluster size and intensity. The Cact-positive zone, however, was not affected (***Figure 6A, B***). We also examined the SSR-localized complex of Pll, Dl and Cact that regulate GluRIIA abundance at the PSD (***Heckscher et al., 2007***). Replacing the wild-type allele with respective pll[25], dl[H], or cact[13] in leon[1]/19-2 suppressed the intensity and size of GluRIIA clusters but had no effect on the thickness of Cact- or Dlg-positive zones (***Figure 6A–D***). These results suggest that dPak and the Dl/Pll/Cact complex may mediate more specifically the expansion of GlrIIA clusters in the *leon* mutant.

To further confirm that reduced expressions of these postsynaptic proteins would suppress SSR or PSD enlargement, *dlgRNAi, dpakRNAi, dlRNAi, cactRNAi,* and *pllRNAi* that have been used previously (***Dent et al., 2015***; ***Sun and Irvine, 2011***; ***Zhou et al., 2015***) or shown in this study (***Figure 6—figure supplement 1A***) to effectively knockdown respective gene expression were introduced into leon[1]/19-2. To show the effect on the membranous SSR, muscle-expressed mCD8GFP that is enriched in the SSR membrane was used (***Figure 6—figure supplement 1B***). Expansion of mCD8GFP areas and enhancement of the signals were also detected in leon[1]/19-2, consistent with the increase of SSR (***Figure 6—figure supplement 1C,D***). Similarly, *dlg* knockdown suppressed mCD8GFP-enriched areas in leon[1]/19-2. However, the expanded PSD-localized GulRIIA clusters remained the same. Instead, *pll, dl* or *cact* knockdown suppressed the intensity and size of GluRIIA clusters in leon[1]/19-2 but had no effect on the expanded mCD8GFP area (***Figure 6—figure supplement 1C,D***). Interestingly, *dpak* knockdown in leon[1]/19-2 suppressed both mCD8GFP areas and GluRIIA clusters (***Figure 6—figure supplement 1C,D***), consistent with its role in the formation

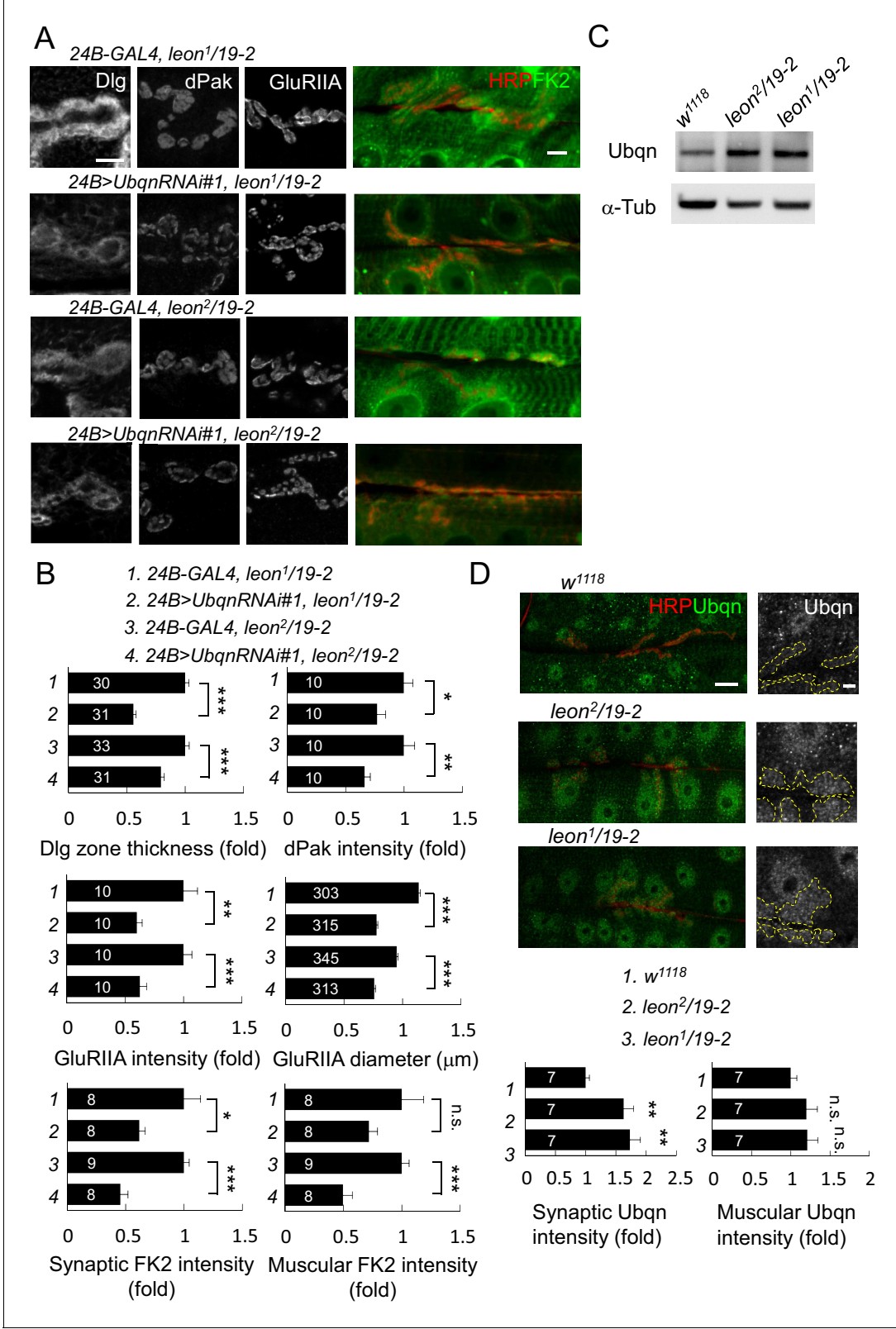

**Figure 7.** Ubqn-dependent postsynaptic expansion and ubiquitin homeostasis defects in *leon* mutants. (**A**) Immunostaining images show suppression of Dlg, dPak and GluRIIA expansions, and reduction of FK2 intensity in *leon[1]/19-2* and *leon[2]/19-2* by *24B-GAL4*-driven postsynaptic expression of *UbqnRNAi#1*. Scale bars: left, 5 μm, and right, 10 μm. (**B**) Bar graphs show means ± SEM of Dlg-positive zone thickness, dPak intensity, GluRIIA intensity
*Figure 7 continued on next page*

of both SSR and GluRIIA clusters (*Albin and Davis, 2004*). Thus, these genetic suppressions support that accumulations of postsynaptic proteins at the SSR or PSD could mediate the expansion of post-synaptic specializations in *leon* mutants.

We then addressed whether the accumulations of postsynaptic proteins at the SSR or PSD are accompanied with increases in the total protein levels. Western blots for examining Dlg, dPak and Cact protein levels were performed in isolated body-wall muscles of wild-type and *leon* mutants. Quantification of the protein levels showed that Dlg, dPak and Cact were increased in *leon$^1$/19-2* (1.19 ± 0.01, 1.3 ± 0.06, and 1.57 ± 0.28 folds, respectively), which were not significantly altered in *leon$^2$/19-2* (0.97 ± 0.06, 0.93 ± 0.09, and 1.13 ± 0.13 folds, respectively). Thus, the increases in the protein levels, as well as other mechanisms, could contribute to postsynaptic accumulations of Dlg, dPak and Cact in *leon* mutants.

## Suppression of *leon* mutant phenotypes by reducing Ubqn levels

In *leon* mutants, ubiquitinated substrates accumulate (*Figure 2A*) while the enzymatic activities of the proteasome remain intact (*Wang et al., 2014*), suggesting that ubiquitinated substrates fail to be transported for proteasomal degradation. We examined the involvement of the ubiquitin-like (UBL) and ubiquitin-association (UBA) domain proteins (UBL-UBA) that function as ubiquitin receptors to bind and shuttle ubiquitinated substrates to the proteasome for degradation. The *Drosophila* genome encodes three UBA-UBL ubiquitin receptors, Rad23, Ddi1 and Ubqn/Dsk2. We then tested whether any of the UBL-UBA proteins contribute to postsynaptic defects in *leon* mutants. RNAi transgenes for knocking down *Ubqn, Rad23* or *Ddi1* were effective in suppressing respective gene expression (*Figure 7—figure supplement 1A*), and were introduced into *leon* mutants to test their suppression of *leon* mutant phenotypes. Interestingly, postsynaptic *Ubqn* knockdown suppressed the size or intensity of Dlg-positive zones, dPak patches and GluRIIA clusters in both *leon$^2$/19-2* and *leon$^1$/19-2* (*Figure 7A,B* and *Figure 7—figure supplement 1B,C*). In contrast, reductions in *Rad23* or *Ddi1* expression showed no obvious alteration of *leon* mutant phenotypes, except for a slight enhancement of Dlg-positive zones in *leon$^2$/19-2* by *Rad23RNAi* (*Figure 7—figure supplement 1B, C*). These results suggest that Ubqn plays a prominent role in mediating postsynaptic phenotypes in *leon* mutants. We then investigated whether *Ubqn* knockdown has any effect on the defective ubiquitin homeostasis in *leon* mutant postsynapses. The FK2 immunostaining intensity, while increased in *leon* mutants, was dramatically reduced in muscles and synapses in *Ubqn* knockdown (*Figure 7A,B*). Therefore, the UBL-UBA protein Ubqn mediates ubiquitin homeostasis defects and postsynaptic defects in *leon* mutants.

Given the reduction in *Ubqn* suppressed *leon* mutant phenotypes, we examined whether Ubqn expression is altered in *leon* mutants. As expected, Ubqn protein levels on Western blots showed increases to more than two folds in both *leon* mutants (*Figure 7C*, *leon$^2$/19-2*: 2.2 ± 0.54 folds and *leon$^1$/19-2*: 2.03 ± 0.35 folds). Immunostaining for Ubqn revealed ubiquitous expression at NMJs and in muscles, with enrichment in nuclei (*Figure 7D*). In *leon* mutants, however, postsynaptic Ubqn immunostaining signals were highly elevated at the SSR (>1.5 folds) and only weakly in other regions (1.2 folds in muscle). These results are consistent with that the increase in the Ubqn level in *leon* mutants could mediate expansions of Dlg-positive zones, dPak patches and GluRIIA clusters in *leon* mutants.

## Ubqn induces and associates with ubiquitinated postsynaptic proteins

We then examined the role of Ubqn in postsynaptic development. The *Flag-Ubqn* transgene expressed by *24B-GAL4* in postsynapses caused expansions of Dlg-positive zones and GluRIIA clusters (*Figure 8A,B*), suggesting that elevated Ubqn levels could promote postsynaptic protein accumulation. However, Ubqn depletion by *UbqnRNAi* knockdown had no effect on these postsynaptic proteins (*Figure 8A,B*). As a ubiquitin receptor, Ubqn could associate with free ubiquitin chains, ubiquitinated substrates, or both in *leon* mutants. Immunoprecipitates of Flag-Ubqn probed by ubiquitin antibodies in Western blots displayed only smearing ubiquitin signals in high molecular weights, which were more prominent in *leon* mutants (*Figure 8C*, left panels). The ubiquitin signals represent Ubqn-associated, ubiquitinated substrates rather than ubiquitinated Ubqn, as SDS treatment of the Flag-Ubqn immunoprecipitates largely depleted the associated signals (*Figure 8C*, right panels).

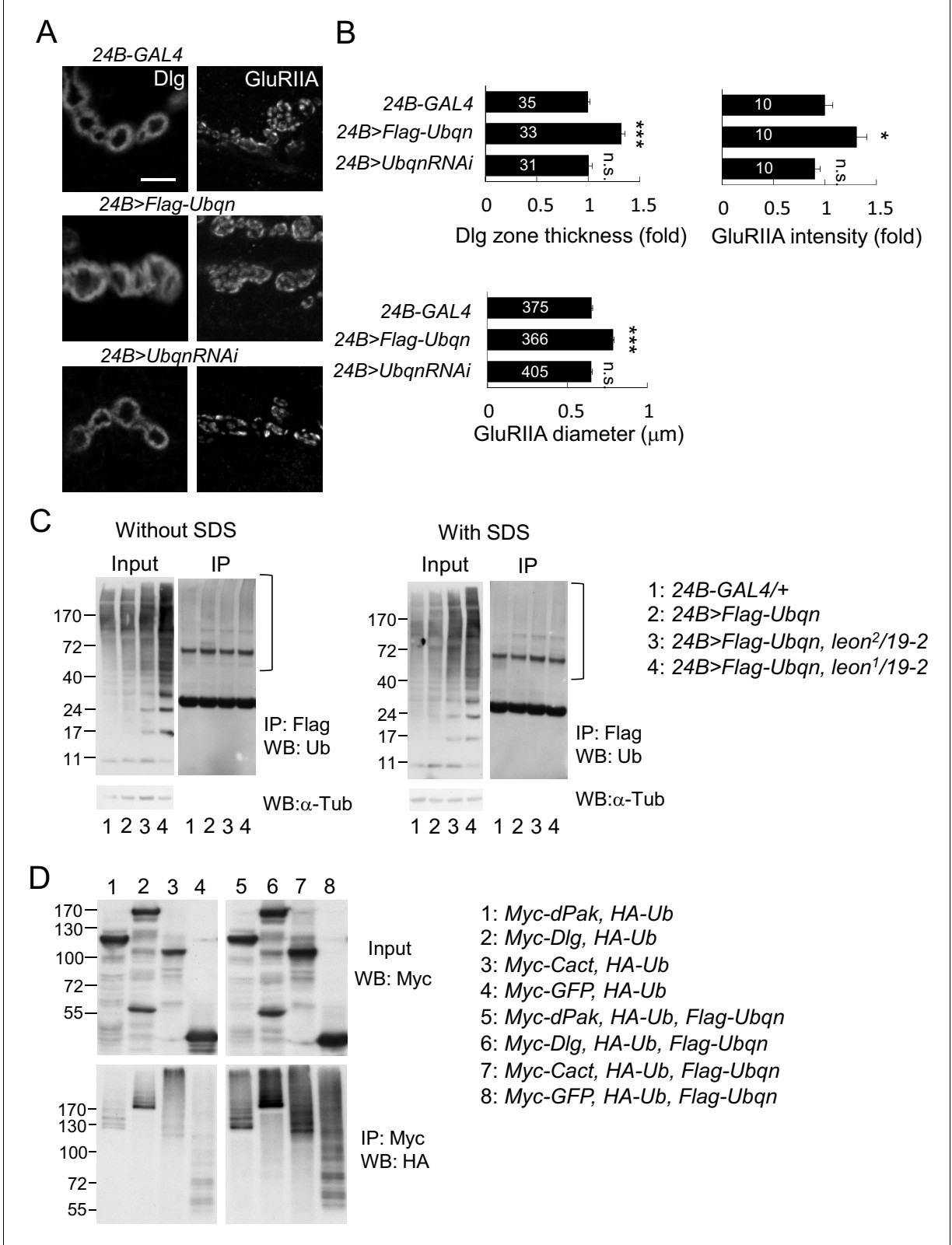

**Figure 8.** Ubqn promotes postsynaptic expansion and associates with ubiquitinated substrates. (A) Images show *24B-GAL4*-driven postsynaptic expression of *Flag-Ubqn* and *UbqnRNAi*, with immunostaining for Dlg or GluRIIA. Scale bar, 5 μm. (B) Bar graphs show means ± SEM of Dlg-positive zone thickness and GluRIIA intensity and diameter. All data were compared to *24B-GAL4* control with n.s. indicating no significance, and *** for p<0.001 according to Student's *t* tests. The detail statistic numbers are in ***Supplementary file 1***. (C) Western blots show input and

*Figure 8 continued on next page*

Figure 8 continued

immunoprecipitation of Flag-Ubqn from *24B-GAL4* control and *24B-GAL4*-driven *Flag-Ubqn* expression in wild type, *leon²/19-2* and *leon¹/19-2*. The immunoprecipitates were probed with ubiquitin antibody. Left panels were performed in normal lysis buffer and right panels in lysis buffer containing SDS to disrupt protei association. Brackets indicate the smearing ubiquitin signals. α-Tub as control. (D) Western blots show input probed by Myc antibody (top panel) and immunoprecipitation of Myc-proteins probed by HA antibody. S2 cell transfected with plasmids for expressing HA-Ub and either Myc-dPak, Myc-Dlg, Myc-Cact, or Myc-GFP (lanes 1–4) or further co-transfected with Flag-Ubqn (lanes 5–8).

Taken together, these results indicate that Ubqn associates with ubiquitinated substrates but not free ubiquitin chains.

We then tested whether elevation of Ubqn could associate and stabilize ubiquitinated postsynaptic proteins. Myc-tagged dPak, Dlg, Cact and GFP were separately co-transfected with HA-Ub into *Drosophila* S2 cells. Myc-immunoprecipitates probed by HA antibodies in Western blots displayed smearing signals (*Figure 8D*, lanes 1–4), indicating they are ubiquitinated proteins. Interestingly, upon further co-transfection of Flag-Ubqn, the Myc-precipitates present stronger ubiquitination signals, suggesting that Ubqn enhances ubiquitination of postsynaptic proteins (*Figure 8D*, lanes 5–8). Ubiquitination of GFP was also enhanced, indicating that Ubqn likely recognizes the conjugated ubiquitin chains for binding, rather than protein substrates. In summary, we propose that Ubqn is elevated in *leon* postsynaptic sites to bind and stabilize ubiquitinated proteins, a mechanism that could account for the accumulation of postsynaptic proteins in *leon* mutants.

## Induction of *leon* mutant phenotypes by free ubiquitin chains

In the absence of Leon deubiquitinating activity, free ubiquitin chains also accumulated in the post-synaptic area (*Figure 2C*). Thus, we tested whether free ubiquitin chains could induce phenotypes observed in *leon* mutants. To increase the level of free ubiquitin chains in vivo, the conserved C-terminal residues Gly75 and Gly76 that are essential for substrate conjugation were mutated to Ala. UbAA can be conjugated by the endogenous ubiquitin C-terminus onto the Lys48 residue, forming free ubiquitin chains, but is unable to conjugate onto substrates including endogenous ubiquitin due to the C-terminal mutations. Also, UbAA-induced free ubiquitin chains are resistant to Leon enzymatic activity as the C-terminal AA motif would prevent recognition by USP5 (*Dayal et al., 2009*). As controls, we also generated wild-type *UAS-HA-UbGG* that can conjugate to substrates, and *UAS-HA-UbAA-K48R* in which Lys48 was replaced by Arg, preventing ubiquitin chain formation on Lys48 of UbAA (*Figure 9A*). Larval lysates of *UAS-HA-UbGG*, *UAS-HA-UbAA* and *UAS-HA-UbAA-K48R* driven by *24B-GAL4* were probed with anti-HA antibodies in Western blot analyses (*Figure 9B*). As expected, UbGG produced smearing signals at higher-molecular weights, representing substrate-conjugated ubiquitin chains (lane 2). As expected, expression of UbAA formed ladders of free ubiquitin chains (lane 3). Finally, expression of UbAA-K48R failed to induce higher-molecular weight, smearing signals and free ubiquitin chains signals, except ubiquitin dimers, likely through conjugation of non-K48 residues (lane 4).

To test which types of ubiquitins have impact on postsynaptic proteins, we performed immunostaining for Dlg, dPak and GluRIIA when these Ub transgenes were expressed in muscles. Quantitative analyses showed that postsynaptic expression of UbAA induced the expansion of Dlg-positive zones and GluRIIA clusters, and enhanced the intensity of dPak and GluRIIA clusters at postsynaptic sites (*Figure 9C,D*). These phenotypes, however, were not detected in expression of UbGG or UbAA-K48R. Given that UbAA, but not UbGG or UbAA-K48R, induced the formation of free ubiquitin chains (*Figure 9B*). Thus, these results are consistent with that K48-linked free ubiquitin chains induce higher-levels of postsynaptic proteins at postsynaptic sites.

However, UbAA overexpression failed to fully recapitulate the extreme severity of *leon* phenotypes, partly due to the difficulty of expressing high levels of free ubiquitin chains in vivo. To further correlate the level of free ubiquitin chains and the phenotypic severity, UbAA was overexpressed in *leon²/19-2* and *leon¹/19-2* in which lower or higher levels of free ubiquitin chains accumulated (*Figure 2A*). Quantification of Dlg-positive zones and GluRIIA cluster intensity and size indicated that overexpression of UbAA further enhanced the hypomorphic *leon²/19-2* phenotypes, but was unable to exacerbate *leon¹/19-2* defects (*Figure 9E*). These analyses suggest that the amounts of free ubiquitin chains correlate to the protein levels at postsynaptic sites.

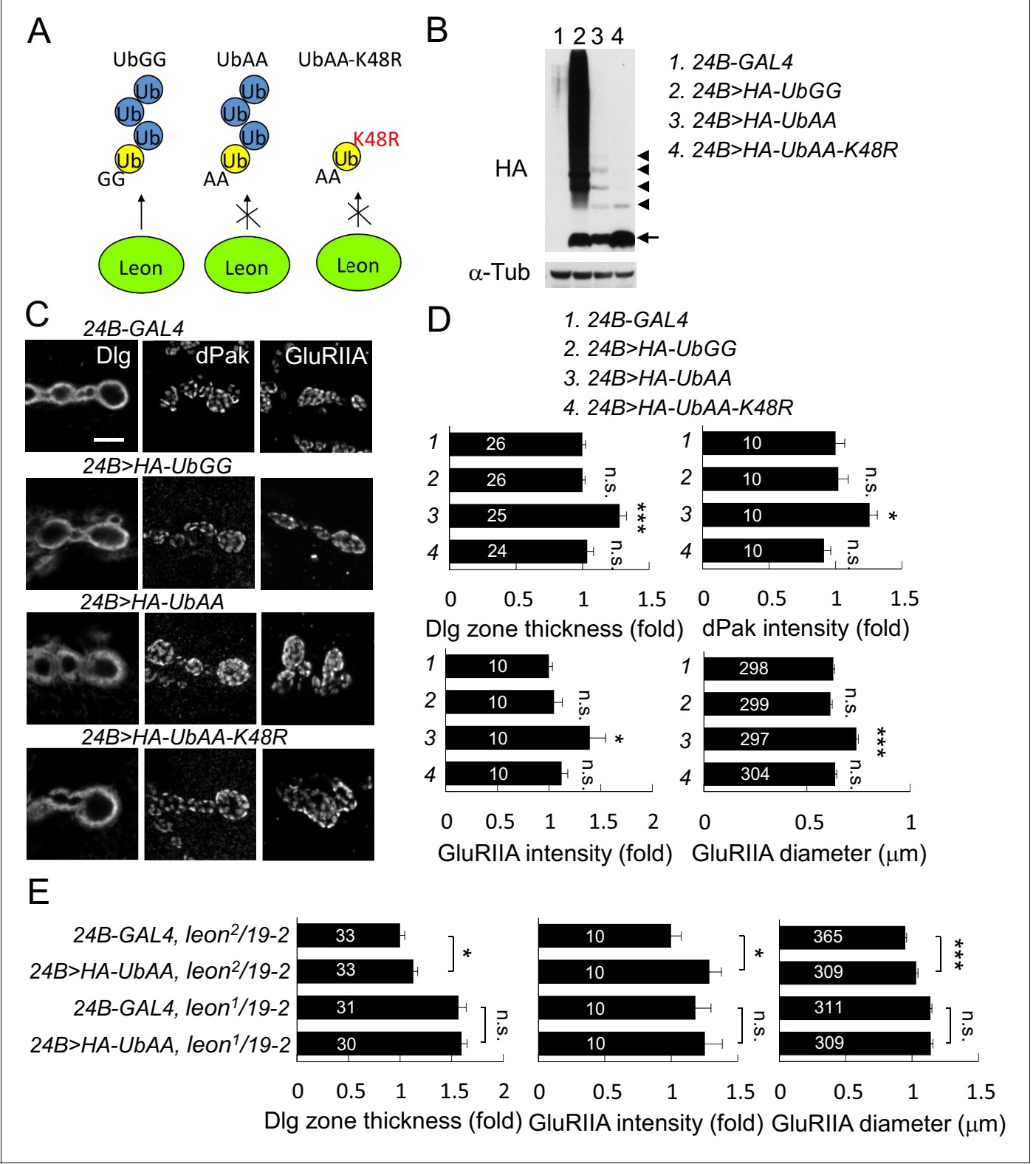

**Figure 9.** Postsynaptic defects induced by K48-linked free ubiquitin chains. (**A**) Diagram shows formation of K48-linked free ubiquitin chains between ectopically expressed UbGG or UbAA (yellow), and endogenous Ub (blue). UbAA-K48R, however, cannot form K48-linked free ubiquitin chains. The free ubiquitin chains initiated by ectopic UbGG, but not UbAA, could be deconjugated by Leon. In addition, UbGG also conjugates to substrates. (**B**) Western blot probed with HA antibody shows HA expression patterns in *24B-GAL4* control (lane 1), *24B>HA-UbGG* (lane 2), *24B>HA-UbAA* (lane 3) and
*Figure 9 continued on next page*

*Figure 9 continued*

*24B>HA-UbAA-K48R* (lane 4). α-Tub as control. Arrow indicates ubiquitin monomer and arrowheads the dimer, trimer, tetramer and pentamer. (**C**) Images show *24B-GAL4*-driven postsynaptic expression of UbGG, UbAA and UbAA-K48R, with immunostaining for Dlg, dPak or GluRIIA. Scale bar, 5 μm. (**D**) Bar graphs show means ± SEM of Dlg-positive zone thickness, dPak intensity, GluRIIA intensity and diameter in *24B-GAL4*-driven expression of UbGG, UbAA and UbAA-K48R. (**E**) Bar graphs show means ± SEM of Dlg-positive zone thickness and GluRIIA intensity and diameter in *24B-GAL4*-driven UbAA expression in *leon²/19-2* or *leon¹/19-2*. All data were compared to controls unless specifically indicated by brackets with n.s. indicating no significance, * for p<0.05, and *** for p<0.001 according to Student's *t* tests. The detail statistic numbers are in ***Supplementary file 1***.

## Recapitulation of *leon* mutant phenotypes by coexpression of free ubiquitin chains and Ubqn

Both levels of free ubiquitin chains and Ubqn were highly elevated in *leon* mutants. Also, when over-expressed, both were able to induce protein accumulation at postsynaptic sites. We then investigated the relationship between free ubiquitin chains and Ubqn in these processes. Postsynaptic expression of UbAA, but not UbGG or UbAA-K48R, slightly induced the Ubqn levels, including synaptic sites and muscles (***Figure 10A,B***). Conversely, overexpression of Ubqn could not induce free ubiquitin chains (***Figure 10C***). As UbAA only induced mild postsynaptic phenotypes accompanying with slight Ubqn elevation, we therefore tested whether co-expression of UbAA and Ubqn could further enhance these phenotypes. This experiment was performed in a *leon* heterozygous background, *+/19–2*, in which postsynapses are normal but more sensitive to the induction of phenotypes. Expression of either UbAA or Ubqn in the sensitive background induced mild enhancement of FK2 intensity in postsynaptic sites. However, combined overexpression of UbAA and Ubqn in the same background caused a large increase in FK2 immunostaining intensity (***Figure 10D,E***). This co-expression also resulted in larger expansion of Cact-positive zones and GluRIIA clusters than expression of either one alone (***Figure 10D,E***). Thus, these data suggest that Ubqn and free ubiquitin chains in *leon* mutants could function together to induce defective ubiquitin homeostasis, and protein accumulations at the postsynaptic site.

## Discussion

At NMJs, postsynaptic SSR membranes are highly convoluted, showing many layers of folded membranes that surround presynaptic boutons. Each bouton contains multiple release sites where neurotransmitters are released from synaptic vesicles and received by receptors localized at the PSD (***Broadie and Richmond, 2002***; ***Collins and DiAntonio, 2007***). Leon negatively regulates these unique postsynaptic specializations through control of postsynaptic protein levels. The Leon function in postsynapses is specific as presynaptic defects were not obvious, and *leon* mutant defects could be rescued mostly by postsynaptic Leon expression. Leon tightly controls postsynaptic specializations throughout larval development. As such dramatic and concomitant expansion of SSRs and PSDs is rarely detected in other reported mutants, our study suggests that Leon likely executes a coordinated program in confining postsynaptic specializations.

The requirement of Leon deubiquitinating activity indicates that Leon functions as a conserved member of the USP5 family to disassemble free ubiquitin chains, thereby maintaining ubiquitin homeostasis in vivo. In *leon* mutants, accumulations of free ubiquitin chains and ubiquitinated substrates were detected (***Figure 2A***). The immunostaining signals by FK2 antibodies were also highly intensified, in particular, at the postsynaptic site (***Figure 2B***). Thus, free ubiquitin chains and ubiquitinated substrates are two major targets that are suppressed by USP5/Leon activity in postsynapses. Postsynaptic expression of UbAA that forms free ubiquitin chains mildly enhanced FK2 intensity and induced higher-levels of SSR- and PSD-localized proteins, similar to but not as dramatic as what was observed in *leon* mutants. In addition to free ubiquitin chains, ubiquitinated substrates also contribute to SSR and PSD expansion. In *leon* mutants, accumulation of ubiquitinated substrates could be attributed to disruption of proteasomal degradation, although the enzymatic activity of the proteasome remains active (***Wang et al., 2014***). We therefore hypothesized that the delivery of ubiquitinated substrates to proteasomal degradation is hindered, leading to their accumulation. In supporting of this idea, we showed that reducing the level of the UBA-UBL ubiquitin shuttling protein Ubqn could suppress SSR- and PSD-localized protein levels in *leon* mutants. Overexpression of

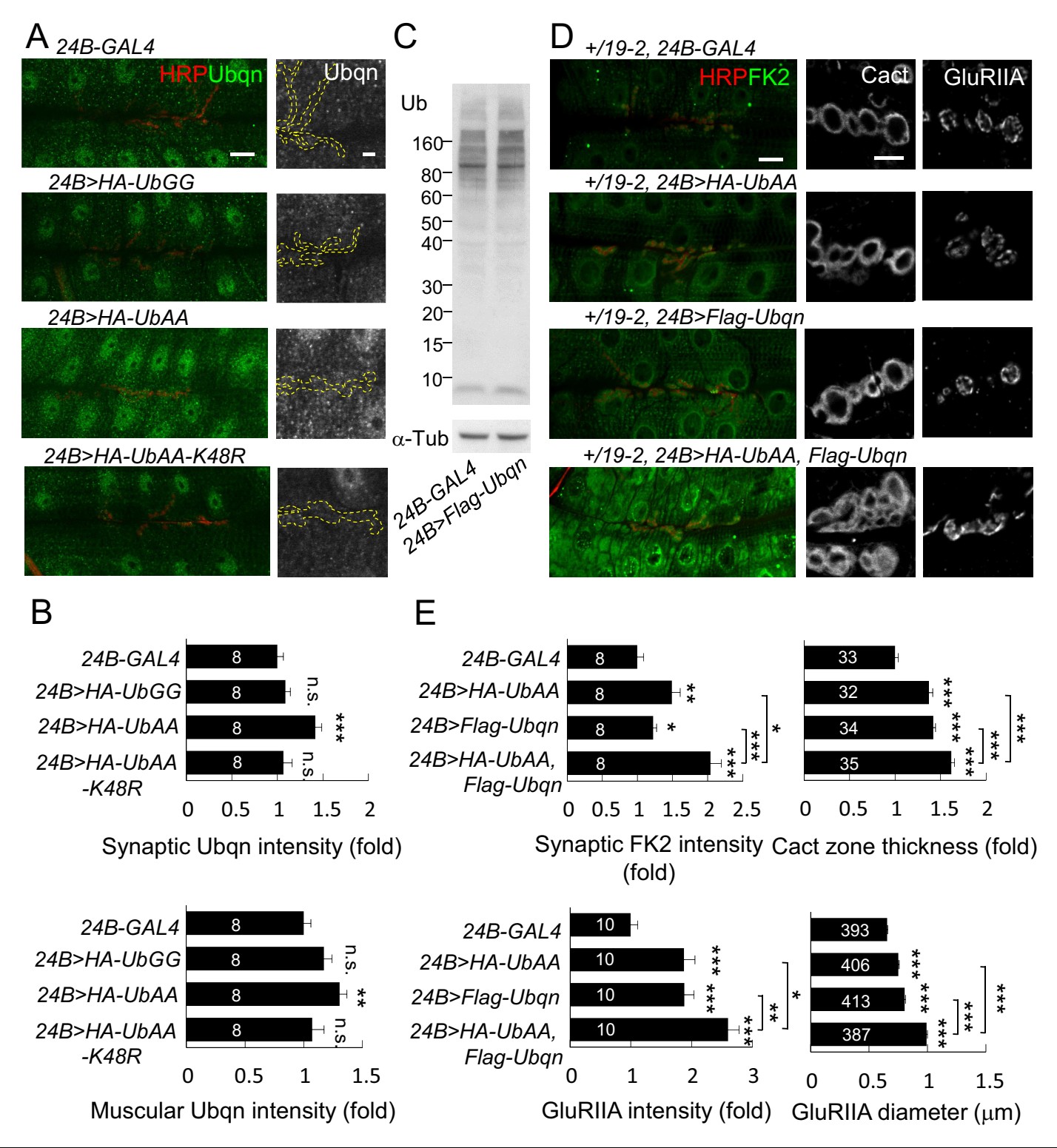

**Figure 10.** Free ubiquitin chains induce Ubqn levels and enhance postsynaptic defects when co-expressed with Ubqn. (**A**) Immunostaining images show NMJs for Ubqn (green) and HRP (red) in *24B-GAL4*-driven postsynaptic expression of UbGG, UbAA and UbAA-K48R. The single Ubqn channel is also shown and yellow dashed lines delineate HRP-positive areas. Scale bars: left, 20 μm and right, 5 μm. (**B**) Bar graphs show means ± SEM of synaptic and muscular Ubqn intensities when UbGG, UbAA or UbAA-K48R was expressed by *24B-GAL4*. (**C**) Western blot probed with Ub antibody shows ubiquitin expression patterns in *24B-GAL4* control and *24B>Flag-Ubqn*. α-Tub as control. (**D**) Images show NMJ immunostaining for FK2 (green, co-

*Figure 10 continued on next page*

Ubqn also caused accumulations of ubiquitinated substrates in postsynapses. The Ubqn protein itself also accumulated at the SSR region, and bound ubiquitinated substrates. Thus, in *leon* mutant postsynapses, higher levels of Ubqn could stall proteasomal degradation and cause accumulation of ubiquitinated substrates in postsynapses (*Figure 11*). Taken together, we propose that accumulated free ubiquitin chains and Ubqn-bound ubiquitinated substrates are two primary causes of *leon* mutant defects.

Interestingly, Ubqn level was also upregulated in UbAA overexpression. Conversely, free ubiquitin chains maintained constant levels when Ubqn was overexpressed. These experiments would suggest

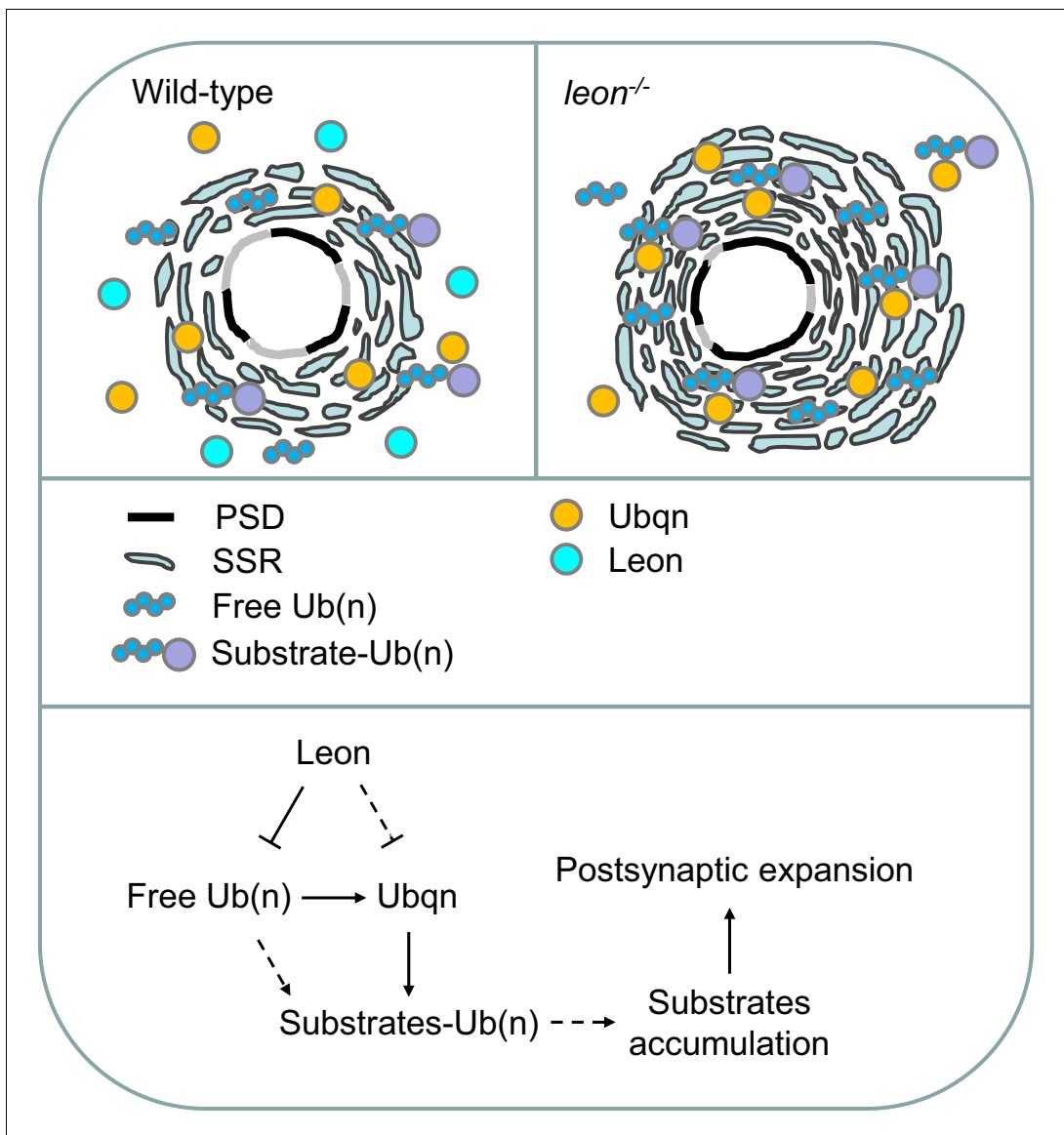

**Figure 11.** Model for Leon in maintaining postsynaptic ubiquitin homeostasis and protein degradation. Schematics show postsynaptic distributions of free and substrate-conjugated ubiquitin chains in wild-type (upper left) and *leon* mutant (upper right) postsynaptic sites. (bottom) The proposed pathway for Leon/Usp5 in postsynapses: Leon downregulates the levels of free ubiquitin chains through the deubiquitinating activity. Accumulation of free ubiquitin chains promotes Ubqn elevation, and Leon may suppress Ubqn levels through alternative pathways (dotted lines). Accumulated Ubqn could bind and stabilize ubiquitinated substrates. However, free ubiquitin chains and Ubqn when both are increased, collaborate to induce more accumulations of ubiquitinated substrates. Finally, accumulated substrates contribute to postsynaptic protein accumulation and SSR and PSD expansions (Solid lines: supported by experiments in this study; dash lines: proposed links).

that accumulation of free ubiquitin chains directly induces higher levels of Ubqn in *leon* mutants, which could be mediated through stabilization of Ubqn or other pathways. The effect of postsynaptic overexpression of UbAA or Ubqn only induced partial *leon* mutant phenotypes, such as in the levels of FK2 and postsynaptic proteins, implying tight control on ubiquitin homeostasis and protein substrate levels. Co-overexpression of UbAA and Ubqn together induced much higher FK2 intensity, and higher postsynaptic protein levels than overexpression of either one alone. Thus, Ubqn and UbAA are unlikely to function in a simple linear pathway, as the overexpression of UbAA would have suggested. Other factors such as ubiquitinated substrates have to be taken into account for the full expressivity of *leon* mutant phenotypes.

With the dramatic expansions of SSR and PSD in *leon* mutants, we assumed many postsynaptic proteins would be elevated in *leon* postsynapses. By examining some representative proteins, we showed that those proteins contribute to *leon* postsynaptic expansions. The SSR-localized Dlg was elevated and its reduction specifically alleviated SSR expansion. PSD-localized dPak was highly elevated near postsynaptic membranes, and dPak accumulation contributes SSR and PSD expansion in *leon* mutants. Two of SSR-localized NF-κB complex, Cact and Dl, were found to accumulate in *leon* mutant SSR. Eliminating one wild-type allele or RNAi knockdown of *cact*, *dl,* or *pll* in *leon* mutants also suppressed the enlargement of GluRIIA clusters, consistent with their roles in regulating GluRIIA cluster abundance. As dramatic increases of protein levels were not detected in Western blots for *leon* mutants, as would be expected from immunostaining of postsynaptic proteins, other mechanisms like subcellular recruitment to postsynapses could be also involved.

How these postsynaptic proteins accumulated at *leon* mutant postsynapses? Ubqn could be the key factor in this process. Ubqn associated with ubiquitinated substrates in *leon* mutants, and could stabilize ubiquitinated dPak, Dlg and Cact (*Figure 8D*). These results suggest that Ubqn could associate with and stabilize ubiquitinated postsynaptic proteins, which contribute to *leon* mutant phenotypes. In particular, Ubqn localized at postsynaptic sites and was enriched locally when *leon* was inactivated, providing the specificity of Leon regulation to postsynaptic specializations (*Figure 11*). Interestingly, while reduction of Ubqn suppressed SSR- and PSD-localized protein levels in *leon* mutants, it enhanced *leon* mutant lethality, indicating the distinct role of Ubqn in postsynapses. Our current model suggests sequential events in *leon* mutant postsynapses, in which elevated free ubiquitin chains induced Ubqn upregulation, Ubqn further stabilized postsynaptic proteins, and accumulated postsynaptic proteins promote SSR and PSD expansion (*Figure 11*). The inability to fully recapitulate *leon* mutant phenotypes could be due to limitation in overexpression or involvement of other factors. Thus, Leon maintains ubiquitin homeostasis through regulation of free ubiquitin chains, Ubqn levels and ubiquitinated substrates for proper development of postsynaptic specializations.

Our model that hinges on postsynaptic Ubqn has resemblances to the pathogenesis of ubiquilin 2-associated ALS (*Deng et al., 2011*; *Ferraiuolo et al., 2011*). Ubiquitin and ubiquilin 2 accumulations are common features in ALS patients, reminiscent of Ubqn and free ubiquitin chain accumulations in *leon* mutants. In addition to ubiquitin homeostatic imbalance and postsynaptic differentiation in *leon* mutants, overexpression of Ubqn causes pupal lethality and modifies TDP-43 toxicity in an ALS model (*Hanson et al., 2010*; *Lipinszki et al., 2011*). Whether ubiquilin 2 accumulation could recruit free ubiquitin chains or ubiquitinated substrates to ubiquitin inclusions and enhance disease progression will be pivotal to understand the pathological mechanisms. Thus, Leon/USP5 holds Ubqn in check and prevents ubiquitin homeostatic imbalance, making *USP5* as a potential candidate disease gene in ubiquitin homeostasis-related diseases.

## Materials and methods

### Fly stocks

*leon[1]*, *leon[2]*, *19–2* mutant alleles, *UAS-Flag-leon* and *UAS-Flag-ED-leon* are described in our previous study (*Wang et al., 2014*). All flies were reared at 25℃.*GFP-leon-GR* was constructed by fusing *GFP* to the ATG codon of *leon* cDNA driven by the genomic sequence between *BtbVII* ATG and *leon* ATG. Transgenic flies carrying *UAS-HA-UbGG*, *UAS-HA-UbAA*, *UAS-HA-UbAA-K48R* and *UAS-Flag-Ubqn* were generated in this study. *dlg[X1-2]* (*Zhang et al., 2007*), *dl[H]*, *cact[13]*and *pll[25]* (*Heckscher et al., 2007*) and *UAS-UbqnRNAi#2* (*Ganguly et al., 2008*) have been described in respective studies. *24B-GAL4* (RRID:BDSC_1767), *D42-GAL4* (RRID:BDSC_8816), *da-GAL4* (RRID:

BDSC_55851), *UAS-mCD8GFP* (RRID:BDSC_5137) and *dpak⁶* (RRID:BDSC_8809) were obtained from Bloomington Drosophila Stock Center, Bloomington, IN. *UAS-leonRNAi* (ID 17567), *UAS-UbqnRNAi#1* (ID 47447), *UAS-Rad23RNAi* (ID 30497), *UAS-Ddi1RNAi* (ID 40512), *UAS-pllRNAi* (ID 103774), *UAS-dlgRNAi* (ID 41136) and *UAS-dpakRNAi* (ID 108937) were from Vienna *Drosophila* Resource Center, Austria. *UAS-cactRNAi* (5848 R-3) was from NIG-FLY, Japan. *UAS-dl-BRNAi* was from Steven A. Wasserman (*Zhou et al., 2015*).

## Immunostaining

NMJ 6/7 phenotypes were analyzed at the A3 segment of late third instar larvae as previously described (*Tsai et al., 2012a*, *2012b*). Larvae were dissected in cold calcium free HL3 saline (70 mM NaCl, 5 mM KCl, 20 mM MgCl₂, 10 mM NaHCO₃, 5 mM trehalose, 115 mM sucrose, and 5 mM HEPES, pH 7.2) and larval body fillets were fixed in 4% paraformaldehyde for 20 min and washed in PBT (0.01% triton-X-100) for 15 min three times. Fixed fillets were incubated with primary antibodies overnight at 4°C, rinsed in PBT three times, and incubated with secondary antibodies for 2 hr at room temperature. Primary antibodies: mouse anti-Dlg (4F3, 1:100; Developmental Studies Hybridoma Bank, DSHB, Iowa City, IA; RRID:AB_528203), mouse anti-Brp (nc82, 1:100; DSHB; RRID:AB_2314868), mouse anti-Synapsin (3C11/SYNORF1, 1:50; DSHB; RRID:AB_528479), mouse anti-CSP (ab49, 1:100; DSHB; RRID:AB_2307345), mouse anti-Futsch (22C10, 1:200; DSHB; RRID:AB_528403), rabbit anti-Cact (1:500; RRID:AB_2314056; [*Reach et al., 1996*]), rabbit anti-Dl rabbit (1:500; RRID: AB_2570310), anti-dPak (1:1000; RRID:AB_2567913; [*Vlachos and Harden, 2011*]), rabbit anti-GluRIIB (1:1000; RRID:AB_2568753; [*Marrus et al., 2004*]), rabbit anti-GluRIIC (1:2500; RRID:AB_2568754; [*Marrus et al., 2004*]), mouse anti-FK2 (1:500; RRID:AB_10541840; Enzo Life Sciences, Farmingdale, NY.) chicken anti-GFP (1:500; RRID:AB_300798; Abcam, UK.), mouse anti-UBQLN2 (1:100; RRID:AB_565683; Abnova, Taiwan), goat anti-HRP conjugated FITC, rabbit anti-HRP conjugated TRITC or Cy5 (RRID:AB_2314647; RRID:AB_2340257; Jackson ImmunoResearch, West Grove, PA) and mouse anti-Leon (1:100) generated against GST-Leon fusion protein (LTK BioLaboratories, Taiwan). Muscles were visualized by FITC-conjugated phalloidin (1:2000; Sigma-Aldrich, St. Louis, Mo), which are not shown in figures. For GluRIIA (mouse, 1:100, RRID:AB_528269; DSHB) immunocytochemistry, larval fillets were fixed in Bouins fixative (Sigma-Aldrich.) for 5 min, followed by the protocol described above. Larval fillets were mounted onto slides with PBS containing 87.5% glycerol and 0.22M 1, 4-diaza-byciclo (2.2.2) octane (Dabco, Sigma-Aldrich.). NMJ images were acquired by confocal Z-stack scanning (Zeiss LSM510, Germany) using 40x water and 100x oil objectives and processed by LSM 5 image examiner and Adobe Photoshop.

## Western blots, Immunoprecipitation and S2 cell transfection

Third instar larva fillets were homogenized in lysis buffer (150 mM NaCl, 5 mM EDTA, 0.5% Triton-X-100, 1% NP-40 and 50 mM Tris-HCl, pH, 7.4) supplemented with protease inhibitor cocktails (Roche, Swiss), 1 mM PMSF, 1 mM DTT and 2 mM Na₃VO₄ and separated in Nupage 4% ~ 12% Bis-Tris gel (Thermo Fisher Scientific, Waltham, MA) for Western blots. For immunoprecipitation, larvae were lysed in lysis buffer or lysis buffer with 0.5% SDS. Lysates with 0.5% SDS buffer were further diluted to final 0.2% SDS for immunoprecipitation. Lysates after incubation with Flag M2 beads (AB_2637089; Sigma-Aldrich) overnight were washed by lysis buffer three times and separated in Nupage 4% ~ 12% Bis-Tris gel for Western blots. *Drosophila* S2 cell line was purchased from Invitrogen (RRID:CVCL_Z232; Invitrogen, Thermo Fisher Scientific) and *Drosophila* S2 cells were cultured at 25°C in Schneider's *Drosophila* medium with 10% heat-inactivated fetal bovine serum (Gibco, Thermo Fisher Scientific). S2 cells were transfected with plasmids by Effectene Transfection Reagent (QIAGEN, Germany). Plasmids include *Ub-GAL4, Myc-dPak, Myc-Dlg, Myc-Cact, Myc-GFP, Flag-Ubqn*, and *HA-Ub*. After 72 hr transfection, S2 cells were lysed in lysis buffer for immunoprecipitation by Myc agarose beads (9E10; Santa Cruz Biotechnology, Dallas, TX). Immunoprecipitated lysates were washed by lysis buffer three times and separated in 8% acrylamide gels. Antibodies against Dlg (1:1000), dPak (1:5000), Cact (1:5000), ubiquitin (P4D1, 1:1000; RRID; AB_628423; Santa Cruz Biotechnology), mouse anti-UBQLN2 (1:100; Abnova), α-Tubulin (1:10000; Sigma-Aldrich), anti-Myc (9E10, 1:1000; RRID:AB_627268; Santa Cruz Biotechnology) and HA-HRP (1:2000; Sigma-Aldrich) were used in Western blots.

## Electron microscopy

Processing and analysis of ultrastructures of synaptic boutons by electron microscope were as described previously (*Tsai et al., 2012b*), with some modification. Larval fillets were dissected in cold calcium-free HL3 saline and fixed in modified Trump's fixative (4% paraformaldehyde/1% glutaraldehyde/0.1 M sodium cacodylate buffer, pH 7.2) and postfixed with 1% aqueous osmium tetroxide/0.1M sodium cacodylate buffer (pH7.2). The dissected muscles 6/7 of A3 segments were stained with 2% uranyl acetate, dehydrated in a graded ethanol series and infiltrated with graded Spurr's series in a microwave (PELCO BioWave Laboratory Microwave System, Ted Pella, Redding, CA). Thin sections (100 nm) were sectioned by ultramicrotome (Leica, Germany.) and further stained with uranyl acetate and lead citrate. Images were acquired by Tecnai G2 Spirit TWIN (FEI Co, Hillsboro, OR.) and a Gatan CCD Camera (794.10 .BP2 MultiScan). TEM data were quantified by MetaMorph V6.3r7 (Molecular Devices, Sunnyvale, CA).

## Quantifications

The thickness of Dlg-, Cact-, or mCD8GFP-positive zone was analyzed in the original confocal Z-stack images and averaged for lengths across the zone defined by eight straight lines radiating from the bouton center. GluRIIA cluster diameters were analyzed in the original confocal Z-stack images, which were scanned with 0.5 µm intervals and covered overall boutons. Isolated GluRIIA cluster with circular shapes in different Z sections were selected to avoid choosing super-imposed ones for scoring their diameters. When elliptic cluster were chosen, the long and short diameters were averaged to reach the average diameters. Brp number and HRP-labeled NMJ areas were analyzed in the original confocal Z-stack images and Brp density was the ratio of Brp number to HRP-labeled NMJ area. Branch length was calculated from HRP labeling. GluRIIA/IIB/IIC and dPak intensities were the summation from all pixels of NMJs, which was normalized to the total HRP intensity at NMJs by Image J. Synaptic FK2 and Ubqn intensities were calculated by mean pixels in the synaptic area normalized to mean HRP pixels in the synaptic area by Image J. Muscular FK2 and Ubqn intensities were averaged from mean pixels of four muscle areas and normalized to mean HRP pixels in the postsynaptic sites by Image J. The intensities were normalized again to the control genotype for presentation in bar graphs.

To estimate expression levels from Western blots, each set of experiments was performed three times independently, and intensities for bands or areas of interest were processed by Image J with normalization to the internal control α-Tub. The areas of interest for mono-ubiquitin, free ubiquitin chains and ubiquitinated substrates are indicated in *Figure 2A*.

## RT-PCR

Larvae for knockdowns of *UAS-UbqnRNAi* (*#1* and *#2*), *UAS-Rad23RNAi*, *UAS-Ddi1RNAi*, *UAS-SyndRNAi* and *UAS-pllRNAi* driven by *da-GAL4* were reared at 25°C. mRNA of indicated genotypes was extracted from third-instar larvae by RNAzol RT (Molecular Research Center, Cincinnati, OH), followed by the reverse transcription of cDNAs by ImProm-II Reverse Transcription System through oligo-dT (Promega, Madison, WI). The mRNA expressions were amplified by PCR using *Ubqn*, *Rad23*, *Ddi1*, *pll* and *Rpl19* specific primers.

## Electrophysiological recordings

For sample preparation, larvae were dissected with the segmental nerve cut near the ventral ganglion in cold modified $Ca^{2+}$-free HL3.1 saline (70 mM NaCl, 5 mM KCl, 10 mM MgCl$_2$, 10 mM NaHCO$_3$, 5 mM trehalose, 115 mM sucrose, 5 mM HEPES, pH 7.2). Preparations were then incubated in modified HL3.1 saline containing 0.6 mM CaCl$_2$ for stimulation, and recordings were taken at room temperature. The two-electrode voltage-clamp was filled with 3 M KCl and impaled in muscle 6 of the A3 segment. One microelectrode (15 ~ 20 MΩ) monitored the muscle membrane potential while the other (5 ~ 8 MΩ) passed electric currents. The muscle membrane potential was clamped at a command value of −60 mV. mEJCs occurring in the background within 100 s were obtained without any stimulation on the segmental nerve. To record an EJC, the segmental nerve was stimulated every 10 s from the cut end that was connected by a suction electrode with 0.1 msec of pulse duration at the voltage two times that of the threshold. Signals were digitized at 50 kHz by a DigiData 1440 interface (Molecular Devices), low-pass filtered at 10 kHz, and saved on an IBM-

compatible PC for analysis. For failure analysis, EJC was evoked in 0.2 mM $[Ca^{2+}]$, and the failure rate was calculated by $\ln(n/N)$, with n the number of failure events, and N the total number of stimuli.

## Acknowledgements

We thank JT Wu, Aaron DiAntonio, Steven A Wasserman, Bloomington Drosophila Stock Center, DGRC, Vienna Drosophila RNAi Center, and DSHB for providing reagents, and SP Lee and SP Tsai for technical support. CTC is supported by grants from Ministry of Science and Technology and Academia Sinica of Taiwan.

## Additional information

### Funding

| Funder | Author |
|---|---|
| Academia Sinica | Chien-Hsiang Wang<br>Yi-Chun Huang<br>Pei-Yi Chen<br>Ying-Ju Cheng<br>Cheng-Ting Chien |
| Ministry of Science and Technology, Taiwan | Chien-Hsiang Wang<br>Yi-Chun Huang<br>Pei-Yi Chen<br>Ying-Ju Cheng<br>Haiwei Pi<br>Cheng-Ting Chien |

The funders had no role in study design, data collection and interpretation, or the decision to submit the work for publication.

### Author contributions

C-HW, Conceptualization, Data curation, Formal analysis, Validation, Investigation, Writing—original draft; Y-CH, Formal analysis, Immunoprecipitation in plasmid tranfected Drosophila S2 cells and Western blots, Figure 2A and 8D are her work; P-YC, Formal analysis, Electrophysiological experiments while preparing resubmitted manuscript, Figure 5-figure supplement 1 is her work; Y-JC, Formal analysis, Transmission electron microscopy, Figure 4 is her work; H-HK, Formal analysis, Electrophysiological experiments in Figure 5; HP, Resources, Supervision, Writing—review and editing; C-TC, Conceptualization, Supervision, Funding acquisition, Project administration, Writing—review and editing

### Author ORCIDs

Cheng-Ting Chien, http://orcid.org/0000-0002-7906-7173

## Additional files

### Supplementary files

• Supplementary file 1. Statistical data for bar graphs shown in figures.

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
