## [Decision Letter]

[Editors’ note: a previous version of this study was rejected after peer review, but the authors submitted for reconsideration. The first decision letter after peer review is shown below.]

Thank you for choosing to send your work, "USP5/Leon deubiquitinase confines postsynaptic differentiation by maintaining ubiquitin homeostasis through Ubiquilin", for consideration at *eLife*. Your article has been reviewed by K VijayRaghavan as Senior editor and three reviewers, one of whom, Hugo Bellen, is a member of the Board of Reviewing Editors. Although the work is of interest, we regret to inform you that the findings at this stage are too preliminary for further consideration at *eLife*. However, we are willing to entertain a resubmission that addresses the reviewers’ comments.

We think that the mechanistic angle including the interpretation of the electrophysiological phenotype will be hard to fix within two months. However, we believe that you should be able to revise the manuscript in a satisfying manner and it should then hopefully be a candidate for *eLife*. We would be happy to re-review such a thoroughly revised version.

*Reviewer #1:*

In the current manuscript, Wang et al. assess the function of the deubiquitinase Leon during *Drosophila* neuromuscular junction (NMJ) development and document that it controls proper postsynaptic differentiation. Leon is the homolog of USP5, a deubiquitinase that functions specifically to disassemble free, substrate-unconjugated polyubiquitin chains to replenish the pool of mono-ubiquitin. Previously, the same group described that loss of *Drosophila* leon is lethal and leads to the accumulation of both free poly-ubiquitin chains, as well as polyubiquitinated proteins. In the current study, the authors describe a role for leon in regulating ubiquitin homeostasis and restricting the growth and development of the postsynaptic density (PSD) and subsynaptic reticulum (SSR): whereas presynaptic components are unaffected, the levels of numerous postsynaptic components (Dlg, GluRIIA/B, dPak, Cact and Dl) are strongly increased. Surprisingly, the observed phenotypes can be significantly rescued by reducing the levels of Ubiquilin, a protein that transports ubiquitinated substrates to the proteasome. In addition, *leon* mutants display increased Ubiquilin levels in *leon* mutants and overexpressing ubiquilin in a wildtype background mimics the phenotypes found in *leon* mutants, albeit to a lesser extent. These data argue for a role of Ubiquilin in mediating PSD and SSR growth. However, free polyubiquitin chains seem to also play a role as overexpressing these chains also shows a milder version of the phenotypes observed in *leon* mutants. Finally, combined overexpression of Ubiquilin and free ubiquitin chains in a wildtype background completely mimics *leon* mutants. Hence, based on the presented data, Leon appears to restrict postsynaptic development by regulating the levels of both Ubiquilin and free polyubiquitin chains.

This is a carefully executed, well-controlled study that consists of an extensive and detailed phenotypic characterization of *leon* mutants. The figures are nicely organized and in most cases very convincingly bring the message across. It is counterintuitive that reducing the levels of a protein that removes polyubiquitinated proteins in a background in which the levels of polyubiquitinated substrates are elevated. However, the genetic experiments that were performed are clean and clearly show that free polyubiquitin chains, in combination with elevated Ubiquilin levels can affect postsynaptic development. How this occurs remains elusive and is not well discussed by the authors. The experiments to tease out the details of the actual mechanism underlying NMJ overgrowth in *leon* mutants are not trivial and may represent a continuation / expansion on this paper. I enjoyed reading this paper and would like to recommend it for publication in *eLife*, if the authors can expand / hypothesize on the consequences of free ubiquitin chains / elevated Ubiquilin levels in the muscle.

Major comment:

– My major comment is a lack of interpretation of the results in the Discussion. At the moment, the Discussion mostly recapitulates the data. How do the authors think Ubiquilin and free ubiquitin chains affect NMJ development?

As the levels of Dlg, Cac, etc are elevated at the earliest stages of NMJ development, it appears that either (a) the activity and/or levels of the master regulator of postsynaptic density formation, which remains unknown, are increased in *leon* mutants or (b) that the individual proteins that are found at the PSD and SSR are directly regulated by Ubiquilin and/or free ubiquitin chains. This is not discussed at the moment, but is essential for the reader to help interpret the consequences of these findings.

How could free ubiquitin chains alter protein levels / activity? Does overexpressing free ubiquitin chains lead to a similar increase in Ubiquilin levels or are these two events independent?

In addition, is the increase in Ubiquilin due to a lack of free ubiquitin monomers? For instance, if the authors overexpress ubiquitin in *leon* mutants, can this rescue the increase in Ubiquilin levels? Does this affect NMJ development?

*Reviewer #2:*

The manuscript reports a very pronounced synaptic phenotype for mutations in Leon, the fly homolog of USP5. Studying the neuromuscular junction, they provide strong evidence that there is an extreme expansion of the folds of the subsynaptic reticulum and a concomittent increase in the postsynaptic presence of proteins associated with that structure. In addition, they show that some proteins found in the postsynaptic density of the NMJ, most notably the receptor subunit GluRIIA, the scaffold protein dlg, and the signaling molecule dPAK are also expanded as is the density by EM. These postsynaptic changes are accompanied by a large change in the gross morpology of the synapse, including a reduction in bouton number. The changes in the nerve structure, however, seem to be an indirect consequence of the role of Leon in the muscle.

The strength of the paper is in the basic description of these phenotypes. The weaknesses of the paper are in the characterization of the electrophysiology and, above all, in the attempt to formulate a mechanistic explanation of the phenotype. There are small technical issues throughout, such as the failure to quantify Western blots and imprecisions in the description (such as equating Cactus or dlg staining with SSR), but where the manuscript really falls apart is from Figure 5 on.

Major Criticisms:

Figure 5: it is not clear to me why they use the heterozygote as the control in panel A. The values for EJC amplitude and quantal content are much lower than the control values in panels B and C. Also, the values for miniEJC amplitude vary greatly from one genetic background to another, hovering around 700 in the D42 and 24B Gal4 lines and more like 900 in the +/19-2 "control". Because this value is the denominator by which they judge quantal content it is possible that the apparent decrease in quantal content is largely due to an increase in mini amplitude that they missed because the +/19-2 value is abnormally high. I would strongly recommend they include *w1118* controls, recorded on the same days, alongside all the measurements for genotypes presented so that one can distinguish effects of the genetic background from those of the mutations. They should use the method of failures to determine quantal content independently since there is good reason to suspect they are misled at present. The increase in quantal size cannot alone explain the decrease in EJC amplitude but might explain a homeostatic decrease in probability of release. In any case, in its current state, the electrophysiology is not interpretable and the final line of that section, which says something vague about correlating with the expansion phenotype, is not really an interpretation and it isn't clear what "correlating" refers to.

Nevertheless, the obvious explanation for the decreased EJC amplitude is that there are far fewer boutons present and therefore fewer release sites and less release. I don't understand why they do not make this point. That miniEJC frequency is normal may reflect a homeostatic compensation, but it seems inadequate to compensate for the sharp drop in bouton #.

Subsection “Impaired electrophysiological properties at *leon* mutant NMJs” paragraph two: do the authors intend mEJC amplitude or EJC amplitude. Did they mean the driver by itself or the driver driving the transgene? This sentence was unclear.

The same paragraph overstates the result. EJC amplitude are not restored to wt levels (and the significance of the change is not tested statistically). What they mean to say is "partially restores" of "improves relative to the mutant" or something like that.

Mechanism: the manuscript does not describe the ubiquitination or a change in ubiquitination for any of the proteins that they claim are misregulated as a result of the leon mutations. They have only overexposed Western blots that are likely to be saturated and no accurate quantification of the levels of the postsynaptic proteins in Figure 6.

Re: Figure 6, one cannot equate Cactus staining with the SSR (see point re Figure 3 also). It might label the entire SSR or it might not, but it would be best just to refer to the width of the cactus immunoreactive zone unless one also is going to do EM to look at the SSR or syndapin staining which might be OK as a surrogate for SSR.

The authors say that knockdown of syndapin has no effect on cactus or GluRIIA expansion but offer no evidence that their RNAi altered syndapin expression. Similarly, they base all the arguments in Figure 6 on the effects of heterozygotes but do not indicate if those heterozygotes changed levels of the encoded protein and over-interpret a failure to alter a parameter as a lack of involvement of that protein in the expansions they observed. Since they do not know quantitatively if leon mutations have altered the levels of that protein, they cannot know if the heterozygous allele they use (a seemingly arbitrary decision) has had any impact on the level of the protein, decreased it back to control levels or not, or rendered it lower than controls. These experiments are not interpretable.

Figure 7 comes the closest to a mechanistic understanding by implicating Ubqn in the pathway. The data are imperfect however since there is no quantification of the claimed increase in Ubqn in the *leon* mutants and the ability of Ubqn loss of function to prevent the expansion of the postsynaptic markers does not really prove that it was directly in the path of the changes. Presumably any protein that was required for the SSR or the PSD would be necessary regardless of whether it was regulated by Leon. Other specific issues for Figure 7 include: In Figure 7, FK2 intensity is measured; is this with background subtraction? Given the big change in overall levels this seems to be necessary, especially since the authors are emphasizing a synaptic decrease, but it is not mentioned in the text/methods. From Figure 7 the authors claim that Ubqn levels are upregulated in *leon* mutants. The change in band size is too subtle to tell, especially with the saturation of both the Ubqn and loading control bands. Either a quantification with several replicates or a shorter exposure blot would be necessary. From Figure 7 the authors claim that Ubqn immunostaining is increased in *leon* mutants, "highly elevated at SSR rather than other regions"; the magnification is too low to tell, and a quantification is necessary to demonstrate the robustness of the phenotype. If quantification cannot be done, please at least show multiple examples. Figure 7—figure supplement 2C: no loading control shown on the blots.

*Reviewer #3:*

Ubiquitin-dependent proteolysis is a complex pathway of protein metabolism implicated in such diverse cellular functions. A late step of the process involves disassembly of the polyubiquitin chains on degraded proteins into ubiquitin monomers. USP5 disassembles branched polyubiquitin chains by a sequential exo mechanism, starting at the proximal end of the chain. Particularly, the enzyme is meant to disassemble free, substrate-unconjugated polyubiquitin chains to replenish the pool of mono-ubiquitin, and thus maintain cellular ubiquitin homeostasis.

The Chien lab recently published a first genetic analysis for USP5 in *Drosophila*, showing that Leon is essential for viability, and that *leon* mutants accumulate both free and substrate-conjugated polyubiquitin chains Proteasomal enzymatic activities were elevated in *leon* mutants, while proteasomal degradation of ubiquitinated substrates was impaired.

Here, the Chien lab executed an analysis of Leon function at the larval NMJ system. Core findings/experiments they present are 1) an expansion of the SSR muscle membrane infoldings and the size of postsynaptic glutamate receptor fields/PSD area due to a lack of muscle Leon function; 2) Western blots and immunostainings of dissected body wall muscles pointing towards increased levels of free and substrate-conjugated polyubiquitin chains; 3) based on TEVC recordings, an impairment of evoked release EJC amplitude and quantal content due to lack of postsynaptic muscle Leon; 4) heterozygous suppression of SSR outgrowth/GluRIIA receptor field outgrowth via dlg /dpak, creating arguments for accumulation of postsynaptic proteins to promote SSR expansion; 5) muscle knockdown of UBA-UBL ubiquitin receptor Ubqn suppressed expansions of Dlg rings and GluRIIA.

Per se, I am impressed by the severity of the phenotypes (SSR expansion, GluR fields) they report, and I think if appropriately revised this manuscript might become a strong candidate for *eLife*. My major concerns are the following:

1) Can they exclude that they are looking at a "heterochronic" phenotype? In the case of dlg mutants, SSR formation is impaired throughout larval development when comparing precisely timed animals. However, as pupa formation is impaired or completely suppressed in dlg mutants, SSR ultimately growths to over a value found in wild type controls. Are their *leon* mutants undergoing pupa formation timely? They should compare precisely timed cohorts to make sure that really the pace of SSR expansion is increased here.

2) Ubqn knockdown they apparently only tested within *leon* mutant background. How does the phenotype look like in wt background? Are they maybe just seeing the addition of two phenotypes here without any specific functional interaction? Anyway, I must confess that I cannot fully follow the mechanistic connection in between Leon and Ubqn.

3) They should try to directly monitor a poly-U species of some relevant postsynaptic protein. This would make their case more convincing.

4) Concerning the diameter of the GluRIIA fields they report: values appear very high. Can they exclude that they are imaging two or more super-imposed receptor fields as one?

5) Somewhat surprisingly but not without example, they find mini amplitudes to be unchanged despite a severe increase of particularly GluRIIA. However, I was surprised to see a down-regulation of evoked release, means that quantal content is severely down here. Concerning the mechanistic basis of their electrophysiological phenotypes, they say: "which correlates with its requirement in postsynaptic differentiation" I am afraid this is insufficient.

6) They state that both free polyubiquitin chains and ubiquitinated substrate proteins were changed in *leon* mutants. Is the availability of monomeric Ubiquitin determining their phenotype? Could they test for this?

7) Why in Figure 2 much more poly-U signal when overexpressing the kinase dead ED Leon?

General comment: the manuscript clearly needs language editing. Moreover, the format of the figures is very diverse, and could be improved by e.g. showing blow ups of the respective GluRIIA phenotypes.

[Editors’ note: what now follows is the decision letter after the authors submitted for further consideration.]

Thank you for resubmitting your work entitled "USP5/Leon deubiquitinase confines postsynaptic growth by maintaining ubiquitin homeostasis through Ubiquilin" for further consideration at *eLife*. Your revised article has been favorably evaluated by K VijayRaghavan (Senior editor), a Reviewing editor, and one reviewer.

The manuscript has been improved but there are some remaining issues that need to be addressed before acceptance, as outlined below. Please make the textual changes suggested by his reviewer..

*Reviewer #1:*

The manuscript is stronger, the electrophysiology more rigorous, and I also think it should be published. The phenotype is big, clear, and the genetics of rescue and RNAi are all very robust. The supplemental table with full statistics is very valuable, as is the inclusion of data from the cd8GFP reporter of the ssr. Some issues can (and should) be improved in the text. My chief concern is that the mechanism behind the phenotypes is not nearly as clear as the authors state. It is fine with me that some things remain poorly explained; but the authors need not try to make it appear as if the very modest changes in certain protein levels are sufficient to explain the phenoptyes or that the changes in bouton number can adequately explain what is happening to quantal content. Ubiquitin homeostasis and protein turn over are very complicated and broadly acting processes and not easily pinned down to the few proteins one looks at. By the end of the Results, a very strong case has been made for the importance of ubiquitination and Ubqn in regulating the development of the postsynapse. The remarks below are to help the authors avoid certain overstatements and premature conclusions in what is otherwise a very interesting and complete manuscript.

1. In the subsection “Increases of postsynaptic proteins in *leon* mutants” and in the Discussion, it is a little awkward to say that each bouton houses tens of synapses (and it isn't always that many) because the term synapse is used sometimes to refer to the whole junction, sometimes to a bouton, and sometimes to each release site. If they say each bouton contains multiple release sites that face discrete receptor clusters the sentence will be less awkward.

2. At the end of the subsection “Impaired electrophysiological properties at *leon* mutant NMJs”: the change in bouton number (which this reviewer suggested as an explanation) doesn't really completely account for the change in the ejc amplitude and quantal content. It may account for some of it, but the ephys parameters are decreased by 50% or more (perhaps 65% for quantal content) and bouton # is only decreased by 20-25%. If the density of release sites per bouton and size of boutons is really unchanged, there remains a mismatch that should still be acknowledged. It is tedious, but it would also help if the authors counted brp puncta in an entire NMJ to get a real value for release sites per NMJ. There may be changes in density, for example, that don't reach statistical significance on their own but, coupled with changes in bouton diameter and number might add up to a larger difference in release sites per NMJ. The bottom line is that it is ok if some of the ephys remains mysterious to explain - but this reviewer did not mean to imply that bouton number alone can actually explain the whole phenotype.

3. There is obviously a disconnect between the very large changes in the SSR and dlg and gluR distributions, and the very small (and in the case of the weaker allele, insignificant) changes in the levels of Pak, dlg, and cact. I think it is fair to guess that many other proteins are increased in order to produce the large changes in the structure of the synapse. In addition, the fact that mutations in those proteins and others can prevent to increase in SSR or GluR markers does not in any way indicate that those proteins are part of the pathway through which leon is acting – it only confirms those proteins are important for normal SSR and GluR development, i.e. they could as easily be part of parallel pathways regulating development as part of a linear pathway.

4. At the end of the subsection “Suppression of *leon* mutant phenotypes by reducing Ubqn levels”: the conclusion regarding the Ubqn increases seen in *leon* mutants is also overstated at this point in the paper. It is certainly likely that this contributes to the other phenotypes and the finding of a large increase in Ubqn in *leon* mutants and the finding that reducing Ubqn can prevent some or all of the phenotypes is important. But to really prove that the change in Ubqn is causative and not merely part of the phenotype, it needs the data in Figure 8 that shows that overexpression of Ubqn in the muscle has the same phenotype as *leon* loss of function. (Ideally, they would also show that simply reducing Ubqn back to wildtype levels with RNAi or a heterozygote would be sufficient to reverse the leon phenotype. Instead they have used RNAi that nearly entirely remove the Ubqn.) The text for Figure 7 should therefore be more circumspect or the big summary of the data for Ubqn in the pathway should wait till after the overexpression of Ubqn is presented in 8A,B.

---

## [Author Response]

[Editors’ note: the author responses to the first round of peer review follow.]

*Reviewer #1:*

*[…] Major comment:*

*– My major comment is a lack of interpretation of the results in the Discussion. At the moment, the Discussion mostly recapitulates the data. How do the authors think Ubiquilin and free ubiquitin chains affect NMJ development?*

*As the levels of Dlg, Cac, etc are elevated at the earliest stages of NMJ development, it appears that either (a) the activity and/or levels of the master regulator of postsynaptic density formation, which remains unknown, are increased in leon mutants or (b) that the individual proteins that are found at the PSD and SSR are directly regulated by Ubiquilin and/or free ubiquitin chains. This is not discussed at the moment, but is essential for the reader to help interpret the consequences of these findings.*

We have revised the discussion on how free ubiquitin chains and Ubqn affect postsynaptic development in *leon* mutants (Discussion section paragraph three on the relationship of Ubqn and free Ub chains, and paragraph five on how SSR and PSD proteins accumulate). In addition, we have performed two additional experiments to address the relationship between free ubiquitin chains, Ubqn and individual SSR or PSD-localized proteins. First, overexpressing free ubiquitin chains also induced Ubqn accumulation (Figure 10, also see response to reviewer#1, point 2). Second, overexpression of Ubqn enhanced/stabilized ubiquitinated dPak, Dlg and Cact (Figure 8). Overall, our model proposes (see Figure 11) that lack of *leon* deubiquitinating activity induces accumulation of free ubiquitin chains, which also contributes to Ubqn upregulation, and both accumulations lead to the accumulations of postsynaptic proteins that induce SSR and PSD growth. Based on this model, the role of Leon/USP5 is to maintain ubiquitin homeostasis, a process critical to postsynaptic growth.

*How could free ubiquitin chains alter protein levels / activity? Does overexpressing free ubiquitin chains lead to a similar increase in Ubiquilin levels or are these two events independent?*

We thank the reviewer for the suggestion of this experiment. We examined whether muscle overexpression of different ubiquitins (UbGG, UbAA and K48RUbAA) could regulate Ubqn levels. The result is that only overexpression of UbAA could upregulate Ubqn, indicating that excessive free ubiquitin chains induce Ubqn accumulation (Figure 10). In contrast, Ubqn overexpression failed to accumulate free ubiquitin chains (Figure 10). Thus, free ubiquitin chains and Ubqn seem to function in a linear pathway upon losing of Leon activity. However, overexpression of UbAA could not fully recapitulate the severity of *leon* phenotypes, which could be that UbAA overexpression hardly reaches the high-level free ubiquitin chains in *leon* mutants, or free ubiquitin chains account partially for Ubqn elevation in *leon* mutants. Ubqn overexpression enhanced the levels of ubiquitinated dPak, Dlg and Cact (Figure 8). The elevated Ubqn could then bind to more ubiquitinated substrates like dPak, Dlg and Cact in postsynapses, leading to their increases in levels and activities. Therefore, this linear part of the pathway from free ubiquitin chains to Ubqn elevation could explain the substrate accumulation, although some other factors could not be excluded.

*In addition, is the increase in Ubiquilin due to a lack of free ubiquitin monomers? For instance, if the authors overexpress ubiquitin in leon mutants, can this rescue the increase in Ubiquilin levels? Does this affect NMJ development?*

We found no evidence to support that the increase of Ubqn is due to a lack of ubiquitin monomer. In *leon* mutants, while Ubqn was increased, the level of ubiquitin monomers was also increased by Western blot analysis (Figure 2). Muscle overexpression of UbGG or K48RUbAA, which produced more ubiquitin monomers (Figure 9), failed to alter Ubqn levels (Figure 10). Thus, the increase of Ubqn is unlikely caused by the lack of ubiquitin monomers. Therefore, we did not perform the rescue experiment of overexpression of UbGG or K48RUbAA in *leon* mutants to suppress Ubqn levels and to affect NMJ development.

*Reviewer #2:*

*[…] Major Criticisms:*

*Figure 5: it is not clear to me why they use the heterozygote as the control in panel A. The values for EJC amplitude and quantal content are much lower than the control values in panels B and C. Also, the values for miniEJC amplitude vary greatly from one genetic background to another, hovering around 700 in the D42 and 24B Gal4 lines and more like 900 in the +/19-2 "control". Because this value is the denominator by which they judge quantal content it is possible that the apparent decrease in quantal content is largely due to an increase in mini amplitude that they missed because the +/19-2 value is abnormally high. I would strongly recommend they include w1118 controls, recorded on the same days, alongside all the measurements for genotypes presented so that one can distinguish effects of the genetic background from those of the mutations. They should use the method of failures to determine quantal content independently since there is good reason to suspect they are misled at present.*

We respond to this comment with three points below:

1)The reason to perform electrophysiology recording with the heterozygous mutant *+/19-2* as the control is to avoid possible background mutations that might arise from the deletion allele. However, we also understand the reviewer’s concern that *+/19-2* could have presented a phenotype in comparison to *w1118*. We therefore preformed new mEJC recording for *w1118, +/19-2* and *leon^[1]^/19-2* and recorded three different genotypes on the same days (Indeed, all our electrophysiological recordings are always done in the same days for different genotypes). The mEJCs of *w1118* and *+/19-2* show little difference (689 v.s. 709 pA). The mEJC of *leon^[1]^/19-2* showed some increases to *w1118* and *+/19-2,* although they are not statistical significant (p=0.3 and 0.39, respectively, by Student’s *t* test) (Figure 5—figure supplement 1). Taken together with data in Figure 5, it seems that mEJCs of *leon^1^/19-2* is likely slightly larger than controls.

2) However, whether the difference in mEJCs is statistical significant or not depends on the genetic background. When performed with GAL4 driver backgrounds (Figure 5), the increases are statistical significant, while without the GAL4 driver background, the increases are not statistical significant by comparison to *w1118* and *+/19-2* (Figure 5 and above). Therefore, we always perform electrophysiological recording with respective GAL4 control in rescue experiments.

3) The slight increase in mEJCs, whether it is statistically significance or not, could not account for the large reduction in the quantal content, which is mainly due to the reduction in EJCs. As suggested by the reviewer, we also preformed failure analysis in *w1118, +/19-2* and *leon^[1]^/19-2* and showed that the quantal content was dramatically decreased in *leon^[1]^/19-2* (Figure 5—figure supplement 1). Thus, the quantal content in *leon^[1]^/19-2* is reduced as compared to both *w1118* and *+/19-2* with studies from two different measurements. The result is described in subsection “Impaired electrophysiological properties at *leon* mutant NMJs “.

*The increase in quantal size cannot alone explain the decrease in EJC amplitude but might explain a homeostatic decrease in probability of release. In any case, in its current state, the electrophysiology is not interpretable and the final line of that section, which says something vague about correlating with the expansion phenotype, is not really an interpretation and it isn't clear what "correlating" refers to.*

*Nevertheless, the obvious explanation for the decreased EJC amplitude is that there are far fewer boutons present and therefore fewer release sites and less release. I don't understand why they do not make this point. That miniEJC frequency is normal may reflect a homeostatic compensation, but it seems inadequate to compensate for the sharp drop in bouton #.*

Due to severe developmental defects in *leon* mutant synapses, we agree that the electrophysiological results are difficult to reconcile in this manuscript. We thank the reviewer for pointing out one better explanation. We rewrote the sentence in the manuscript to describe that fewer boutons might be correlative to lower EJCs and quantal contents detected in *leon* mutants (subsection “Impaired electrophysiological properties at *leon* mutant NMJs”).

*Subsection “Impaired electrophysiological properties at leon mutant NMJs” paragraph two: do the authors intend mEJC amplitude or EJC amplitude. Did they mean the driver by itself or the driver driving the transgene? This sentence was unclear.*

This sentence was really unclear.Actually, we reorganized this section.The sentence is shown as “The mEJC in *leon* mutants carrying *D42-GAL4* or *24B-GAL4* was significantly larger than that in respective GAL4 driver control, suggesting the mEJC increase in *leon* mutants might be sensitive to the variation in genetic backgrounds (Figure 5).”

*The same paragraph overstates the result. EJC amplitude are not restored to wt levels (and the significance of the change is not tested statistically). What they mean to say is "partially restores" of "improves relative to the mutant" or something like that.*

We agree it is a partial rescue andrewrote the sentence as “Postsynaptic Leon expression by *24B-GAL4* partially restored the EJC amplitude and the quantal content in *leon^[1]^/19-2* (Figure 5).”. The statistical test for partial recue to *24B-GAL4* control is now indicated in Figure 5.

*Mechanism: the manuscript does not describe the ubiquitination or a change in ubiquitination for any of the proteins that they claim are misregulated as a result of the leon mutations. They have only overexposed Western blots that are likely to be saturated and no accurate quantification of the levels of the postsynaptic proteins in Figure 6.*

To quantify protein levels for Dlg, dPak, and Cact, immunoblot intensities from three independent Western blots were averaged. The result suggests that the protein levels are significantly increased in the *leon^[1]^/19-2* mutant and remained similar in *leon^2^/19-2* (Figure 6, the quantifications are described in subsection “Suppression of leon mutant phenotypes by reducing postsynaptic proteins”). Therefore, we propose that both increase of protein levels and localization to postsynaptic sites are two major regulations. We have also changed images with shorter exposure for Figure 6.

Regarding ubiquitination, the ubiquitinated proteins are difficult to detect in Western blot for body wall muscle extracts. We have tried to overexpress tagged transgenes for Ub and Dlg or dPak in muscles and still had difficulty to detect any changes in *leon* mutants. One possibility is that the pools of proteins in postsynaptic sites are not sufficient for detecting any change. We also tried to knockdown of Leon in S2 cells by dsRNA for examing Dlg, dPak and Cact ubiquitination. While 90% of Leon was depleted in S2 cells, such knockdown efficiency is not sufficient to disrupt ubiquitin homeostasis in S2 cells, as free ubiquitin chains and ubiquitinated substrates could not be detected. One alternative experiment that we have performed is the detection of protein ubiquitination with or without Ubqn in S2 cells (Figure 8). We found that ubiquitinations of dPak, Dlg and Cact were elevated when Ubqn was overexpressed. Thus, Ubqn is the important player in mediating protein accumulation in *leon* mutants. The results are described in a paragraph two of subsection “Ubqn induces and associates with ubiquitinated postsynaptic proteins”, and discussed in paragraph five of the Discussion section.

*Re: Figure 6, one cannot equate Cactus staining with the SSR (see point re Figure 3 also). It might label the entire SSR or it might not, but it would be best just to refer to the width of the cactus immunoreactive zone unless one also is going to do EM to look at the SSR or syndapin staining which might be OK as a surrogate for SSR.*

We agree with the reviewer’s concern, and performing EM to assess the genetic suppression is an unlikely task, considering there are so many genetic assays. We were not able to obtain Syndapin antibodies for the assessment, either. We therefore used the membrane form of GFP that is enriched in SSR to repeat the genetic suppression test (Figure 6—figure supplement 1). Also, we change the term SSR to Cact-positive zones.

*The authors say that knockdown of syndapin has no effect on cactus or GluRIIA expansion but offer no evidence that their RNAi altered syndapin expression.*

The *UAS-SyndRNAi#1*, used in previous submitted manuscript, is efficient in knockdown as shown by RT-PCR (see Figure 12). However, we decide to remove the results about the non-suppression by *Syndapin* knockdown, which provides no extra meaning to the manuscript.

Author response image 1.**DOI:**
http://dx.doi.org/10.7554/eLife.26886.022

*Similarly, they base all the arguments in Figure 6 on the effects of heterozygotes but do not indicate if those heterozygotes changed levels of the encoded protein and over-interpret a failure to alter a parameter as a lack of involvement of that protein in the expansions they observed. Since they do not know quantitatively if leon mutations have altered the levels of that protein, they cannot know if the heterozygous allele they use (a seemingly arbitrary decision) has had any impact on the level of the protein, decreased it back to control levels or not, or rendered it lower than controls. These experiments are not interpretable.*

The alleles we used for *dlg, dpak, pll, dl* and *cact* in Figure 6 and Figure 6—figure supplement 1 are all documented null alleles that have been reported (Albin and Davis, 2004; Heckscher et al., 2007; Zhang et al., 2007). Heterozygous mutants showed a suppression effect on one or the other phenotype (Cact/Dlg or GluRIIA expansion) in *leon* mutants. Thus, whether protein levels are reduced or not, they should have reduced the gene activity in the heterozygous background. Nevertheless, we also tested the RNAi knockdown of *dlg, dpak, pll, dl* and *cact* in muscles of the *leon* mutants, and show the effect on suppression of mCD8GFP-labled SSR or GluRIIA expansions (Figure 6—figure supplement 1). These RNAi lines have been used in other studies (Dent et al., 2015; Sun and Irvine, 2011; Zhou et al., 2015), except *pelle* that was examined in this study (Figure 6—figure supplement 1). The results showed almost the same suppression effects as by heterozygous mutants. In addition, *dpak* knockdown further suppressed mCD8GFP-labled SSR expansion, in consistent with previous study in which dPak regulates GluRIIA abundance and SSR formation (Albin and Davis, 2004).

*Figure 7 comes the closest to a mechanistic understanding by implicating Ubqn in the pathway. The data are imperfect however since there is no quantification of the claimed increase in Ubqn in the leon mutants and the ability of Ubqn loss of function to prevent the expansion of the postsynaptic markers does not really prove that it was directly in the path of the changes. Presumably any protein that was required for the SSR or the PSD would be necessary regardless of whether it was regulated by Leon.*

We quantified three independently performed Western blots and immunostaining of Ubqn (Figure 7, respectively, the quantification results are described in subsection “Suppression of leon mutant phenotypes by reducing Ubqn levels”). Both assays showed that the Ubqn levels were increased in hypomorphic *leon^[2]^/19-2* and null *leon^[1]^/19-2*.We address that Ubqn is in the Leon/USP5 pathway with following data: (1) The Ubqn level increased in *leon* mutants by both Western blots and immunostaining assays (Figure 7), and the increases are more specific to synaptic areas than in overall muscles; (2) Muscle knockdown of *Ubqn* suppressed *leon* mutant phenotypes in SSR and PSD expansions (Figure 7 and Figure 7—figure supplement 1); (3) Muscle overexpression of *Ubqn* caused expansions of Dlg-positive zone and GluRIIA diameter (Figure 8); (4) Ubqn bound and stabilized ubiquitinated proteins that are increased in *leon* mutants (Figure 8). Taken together, Ubqn is a critical player in mediating *leon* mutant phenotypes.

*Other specific issues for Figure 7 include: In Figure 7, FK2 intensity is measured; is this with background subtraction? Given the big change in overall levels this seems to be necessary, especially since the authors are emphasizing a synaptic decrease, but it is not mentioned in the text/methods.*

We think that the FK2 signals are present in both postsynaptic sites as well as in muscle, and both are affected in *Ubqn* knockdown. Therefore, we did not treat the muscle signals as background. We now show quantifications of FK2 intensities for synapses as well as muscles (Figure 7, bottom two panels).

*From Figure 7 the authors claim that Ubqn levels are upregulated in leon mutants. The change in band size is too subtle to tell, especially with the saturation of both the Ubqn and loading control bands. Either a quantification with several replicates or a shorter exposure blot would be necessary.*

We replaced it with a new Western blot image and quantified Ubqn levels (normalized to α-Tub internal control) from three independent experiments (Figure 7, the quantifications are described in subsection “Suppression of leon mutant phenotypes by reducing Ubqn levels”).

*From Figure 7 the authors claim that Ubqn immunostaining is increased in leon mutants, "highly elevated at SSR rather than other regions"; the magnification is too low to tell, and a quantification is necessary to demonstrate the robustness of the phenotype. If quantification cannot be done, please at least show multiple examples.*

We showed higher magnification figures in Ubqn single channel and used dashed lines to outline the Cact-positive regions (not shown in the figure) for quantification. We also quantified the mean intensity of Ubqn in muscles and Cact-positive zones in wild type and *leon* mutants as suggested (Figure 7). The result is also described in subsection “Suppression of leon mutant phenotypes by reducing Ubqn levels”.

*Figure 7—figure supplement 2C: no loading control shown on the blots.*

We have added loading controls (Figure 8).

*Reviewer #3:*

*[…] 1) Can they exclude that they are looking at a "heterochronic" phenotype? In the case of dlg mutants, SSR formation is impaired throughout larval development when comparing precisely timed animals. However, as pupa formation is impaired or completely suppressed in dlg mutants, SSR ultimately growths to over a value found in wild type controls. Are their leon mutants undergoing pupa formation timely? They should compare precisely timed cohorts to make sure that really the pace of SSR expansion is increased here.*

We noticed that the entire larval stage in the *leon^[1]^/19-2* mutant was one day longer than wild type. The expansion of SSR and PSD in the *leon^[1]^/19-2* mutants examined in the late third instar stage may be a consequence of extra growth. We have performed the experiment as suggested by the reviewer to compare SSR and PSD phenotypes between control and *leon* mutants at 72h, 96h, and 120h after egg laying (AEL). Apparently, *leon* mutants presented enlargement of PSD and SSR as early as 72h AEL, as the Cact-positive zone and GluRIIA clusters are significantly larger than controls, and the differences are further enhanced at later stages (Figure 3—figure supplement 2). The description for this result could be found in paragraph four of subsection “Increases of postsynaptic proteins in *leon* mutants”.

*2) Ubqn knockdown they apparently only tested within leon mutant background. How does the phenotype look like in wt background? Are they maybe just seeing the addition of two phenotypes here without any specific functional interaction? Anyway, I must confess that I cannot fully follow the mechanistic connection in between Leon and Ubqn.*

Muscle *Ubqn* knockdown in the wildtype background had no effect on SSR-localized Dlg and PSD-localized GluRIIA (Figure 8). Thus, the suppression of *leon* mutant phenotypes by *Ubqn* knockdown are not the summation of two phenotypes. The explanation for the connection between *leon* and *Ubqn* could be found in response to reviewer#1, point 2, and more thoroughly in Discussion (paragraphs three and five). Briefly, elevated free ubiquitin chains in *leon* mutants induced elevation of Ubqn levels. Accumulated Ubqn stabilizes ubiquitinated postsynaptic proteins to cause NMJ defects.

*3) They should try to directly monitor a poly-U species of some relevant postsynaptic protein. This would make their case more convincing.*

It has been difficult to detect ubiquitinated dPak, Dlg and Cact in body wall muscle preparation of wildtype or *leon* mutants, even with overexpression of Ub transgene. However, in S2 cells, expression of Ubqn stabilizes ubiquitinated dPak, Dlg and Cact, as compared to those without Ubqn overexpression. These results were shown in Figure 8 and described in paragraph two of subsection “Ubqn induces and associates with ubiquitinated postsynaptic proteins”. These results are consistent with that accumulation of Ubqn, such as in *leon* mutants, stabilizes ubiquitinated dPak, Dlg and Cact. Also see response to reviewer#2, point 5.

*4) Concerning the diameter of the GluRIIA fields they report: values appear very high. Can they exclude that they are imaging two or more super-imposed receptor fields as one?*

In our analysis,the diameter of GluRIIA clusters is about 0.63 µm in wild type third instar larvae, which is comparable to GluR cluster diameters in a previous report from Graeme Davis’s group, showing that the GluRIIC diameter is about 0.78 µm in third instar larvae and the GluRIIA diameter is about 0.52 µm in second instar larvae (Pielage et al., 2006). In our study, we were very cautious to select “isolated” GluRIIA puncta in both mutants and wild type. The methodology is described in Materials and methods (“Quantifications”). Briefly,Z stacks of GluRIIA images were scanned with 0.5 µm intervals. Isolated puncta in different Z sections were selected for scoring, thus avoiding super-imposed ones.

*5) Somewhat surprisingly but not without example, they find mini amplitudes to be unchanged despite a severe increase of particularly GluRIIA. However, I was surprised to see a down-regulation of evoked release, means that quantal content is severely down here. Concerning the mechanistic basis of their electrophysiological phenotypes, they say: "which correlates with its requirement in postsynaptic differentiation" I am afraid this is insufficient.*

We think it is not easy to explain the down-regulation of evoked release through postsynaptic structural defects. One possible explanation is that the overall bouton number was decreased in *leon* mutants, leading to the decreased evoked response (as suggested by reviewer 2). Thus, the description is rewritten as “As postsynaptic expression of Leon also suppressed bouton reduction in the *leon* mutant (Figure 1), the reduction of the EJC amplitude and the quantal content could be attributed to the reduction of the bouton and hence the release sites in *leon* mutants”.

*6) They state that both free polyubiquitin chains and ubiquitinated substrate proteins were changed in leon mutants. Is the availability of monomeric Ubiquitin determining their phenotype? Could they test for this?*

Muscle expression of K48R-UbAA, which mimics the accumulation of mono-ubiquitin but not K48-linked polyubiquitin chains, generates large amounts of ubiquitin monomer (Figure 9, lane 4). However, it cannot promote SSR and PSD expansion. Instead, UbAA that could form free ubiquitin chains (lane 3) promotes SSR and PSD expansion. While ubiquitin monomers were increased in *leon* mutants (Figure 2), elevated ubiquitin monomers seem have no contribution to SSR and PSD expansion.

*7) Why in Figure 2 much more poly-U signal when overexpressing the kinase dead ED Leon?*

We think that ED-Leon could associate with polyubiquitin chains but fail to disassemble polyubiquitin chains due to inactivation of the enzymatic activity. It is possible that binding to ED-Leon sequesters ubiquitinated substrates from degradation and leads to their accumulation.

*General comment: the manuscript clearly needs language editing. Moreover, the format of the figures is very diverse, and could be improved by e.g. showing blow ups of the respective GluRIIA phenotypes.*

The revised manuscript has been reviewed by an in-house English editor. Also, in the revised manuscript, GluRIIA images are enlarged, and representative enlarged images are shown in Figure 3 (right panels). We also made figure and bar graph format consistent throughout the text.

[Editors’ note: the author responses to the second round of peer review follow.]

*The manuscript is stronger, the electrophysiology more rigorous, and I also think it should be published. The phenotype is big, clear, and the genetics of rescue and RNAi are all very robust. The supplemental table with full statistics is very valuable, as is the inclusion of data from the cd8GFP reporter of the ssr. Some issues can (and should) be improved in the text. My chief concern is that the mechanism behind the phenotypes is not nearly as clear as the authors state. It is fine with me that some things remain poorly explained; but the authors need not try to make it appear as if the very modest changes in certain protein levels are sufficient to explain the phenoptyes or that the changes in bouton number can adequately explain what is happening to quantal content. Ubiquitin homeostasis and protein turn over are very complicated and broadly acting processes and not easily pinned down to the few proteins one looks at. By the end of the Results, a very strong case has been made for the importance of ubiquitination and Ubqn in regulating the development of the postsynapse. The remarks below are to help the authors avoid certain overstatements and premature conclusions in what is otherwise a very interesting and complete manuscript.*

*1. In the subsection “Increases of postsynaptic proteins in leon mutants” and in the Discussion, it is a little awkward to say that each bouton houses tens of synapses (and it isn't always that many) because the term synapse is used sometimes to refer to the whole junction, sometimes to a bouton, and sometimes to each release site. If they say each bouton contains multiple release sites that face discrete receptor clusters the sentence will be less awkward.*

Following the reviewer’s suggestion, we modified the sentences as “At NMJs, each bouton contains multiple release sites paired with discrete receptor clusters that…” and “Each bouton contains multiple release sites where neurotransmitters…”.

*2. At the end of the subsection “Impaired electrophysiological properties at leon mutant NMJs”: the change in bouton number (which this reviewer suggested as an explanation) doesn't really completely account for the change in the ejc amplitude and quantal content. It may account for some of it, but the ephys parameters are decreased by 50% or more (perhaps 65% for quantal content) and bouton # is only decreased by 20-25%. If the density of release sites per bouton and size of boutons is really unchanged, there remains a mismatch that should still be acknowledged. It is tedious, but it would also help if the authors counted brp puncta in an entire NMJ to get a real value for release sites per NMJ. There may be changes in density, for example, that don't reach statistical significance on their own but, coupled with changes in bouton diameter and number might add up to a larger difference in release sites per NMJ. The bottom line is that it is ok if some of the ephys remains mysterious to explain - but this reviewer did not mean to imply that bouton number alone can actually explain the whole phenotype.*

We quantified the Brp number per NMJ: *w^1118^*, 731.6 ± 31.6; *leon^2^/19-2*, 631.9 ± 28.1; *leon^1^/19-2*, 619.5 ± 31.1; n = 10 for all genotypes; Statistics by Student’s t test showed that both leon mutants were p < 0.05 in comparison to *w^1118^*. These reductions of Brp numbers are about 15%, and are close to the 20% reduction in the bouton numbers in *leon* mutants. As EJC and quantal content were decreased by 50%, we think reduction of bouton and the releasing sites contribute partly to the reduction of EJC and quantal content. We present this quantification and modified the description accordingly.

*3. There is obviously a disconnect between the very large changes in the SSR and dlg and gluR distributions, and the very small (and in the case of the weaker allele, insignificant) changes in the levels of Pak, dlg, and cact. I think it is fair to guess that many other proteins are increased in order to produce the large changes in the structure of the synapse. In addition, the fact that mutations in those proteins and others can prevent to increase in SSR or GluR markers does not in any way indicate that those proteins are part of the pathway through which leon is acting – it only confirms those proteins are important for normal SSR and GluR development, i.e. they could as easily be part of parallel pathways regulating development as part of a linear pathway.*

We agree that many proteins are altered in *leon* mutants, and we are only able to examine a handful of them. The slight but not dramatic increases in protein levels of dPak, Dlg and Cact as shown by Western blots could explain at least partly the protein accumulation in postsynaptic specializations. Other mechanisms such as protein recruitment could be also involved. We modified the texts to describe this idea in the Results and Discussion. Regarding whether these proteins are part of the Leon pathway or not, we have not implied that Leon is in direct regulation of these proteins. Instead, we have included Ubqn in the model to explain how these proteins might be stabilized and contribute to postsynaptic structural expansions (Discussion, and Figure 11 for a schematic model).

4. At the end of the subsection “Suppression of leon mutant phenotypes by reducing Ubqn levels”: the conclusion regarding the Ubqn increases seen in leon mutants is also overstated at this point in the paper. It is certainly likely that this contributes to the other phenotypes and the finding of a large increase in Ubqn in leon mutants and the finding that reducing Ubqn can prevent some or all of the phenotypes is important. But to really prove that the change in Ubqn is causative and not merely part of the phenotype, it needs the data in Figure 8 that shows that overexpression of Ubqn in the muscle has the same phenotype as leon loss of function. (Ideally, they would also show that simply reducing Ubqn back to wildtype levels with RNAi or a heterozygote would be sufficient to reverse the leon phenotype. Instead they have used RNAi that nearly entirely remove the Ubqn.) The text for Figure 7 should therefore be more circumspect or the big summary of the data for Ubqn in the pathway should wait till after the overexpression of Ubqn is presented in 8A,B.

We modified the conclusion for Fig. 7. This modified text was shown in Lines 431-433, as “These results are consistent with that the increase in the Ubqn level in *leon* mutants could mediate expansions of Dlg-positive zones, dPak patches and GluRIIA clusters in *leon* mutants.”